# GVKF: Gaussian Voxel Kernel Functions for Highly Efficient Surface Reconstruction in Open Scenes

**Gaochao Song**[*] **Chong Cheng**[*] **Hao Wang**[†]
The Hong Kong University of Science and Technology (Guangzhou)
gcsong4@gmail.com
ccheng735@connect.hkust-gz.edu.cn
haowang@hkust-gz.edu.cn

## Abstract

In this paper we present a novel method for efficient and effective 3D surface reconstruction in open scenes. Existing Neural Radiance Fields (NeRF) based works typically require extensive training and rendering time due to the adopted implicit representations. In contrast, 3D Gaussian splatting (3DGS) uses an explicit and discrete representation, hence the reconstructed surface is built by the huge number of Gaussian primitives, which leads to excessive memory consumption and rough surface details in sparse Gaussian areas. To address these issues, we propose Gaussian Voxel Kernel Functions (GVKF), which establish a continuous scene representation based on discrete 3DGS through kernel regression. The GVKF integrates fast 3DGS rasterization and highly effective scene implicit representations, achieving high-fidelity open scene surface reconstruction. Experiments on challenging scene datasets demonstrate the efficiency and effectiveness of our proposed GVKF, featuring with high reconstruction quality, real-time rendering speed, significant savings in storage and training memory consumption. Project page: https://3dagentworld.github.io/gvkf/.

## 1 Introduction

3D surface reconstruction in open scenes holds great significance in various practical applications, such as autonomous driving, virtual reality, urban planning and etc. However, achieving high-fidelity and efficient open scene reconstruction has been a longstanding challenge, due to the trade-off between the rendering quality and the required resources for optimization.

In pursuit of this goal, two predominant approaches are Neural Radiance Fields (NeRF) [27, 40, 1, 11, 19] and 3D Gaussian Splatting (3DGS) [15, 4, 21, 41] based methods. On one hand, NeRF-based implicit representations typically require extensive training and rendering time, which limits the practical use in large-scale scene reconstruction [11, 38, 26, 36]. On the other hand, 3DGS [15] adopts explicit representations, which enables high-quality novel view synthesis while achieving real-time rendering. This makes 3DGS more feasible for efficient scene reconstruction in the applications such as autonomous driving and virtual reality.

Recently, there are studies using 3DGS technology for novel view synthesis and surface reconstruction in street scenes and urban environments [4, 21, 41, 23]. For instance, SuGaR [12] attempts to reconstruct the 3D surfaces based on Gaussian points. However, it has been noted that overly large and sparse Gaussian points can significantly affect the geometric representations of the scene,

---

[*]Both authors contributed equally to this research.
[†]The corresponding author.

38th Conference on Neural Information Processing Systems (NeurIPS 2024).

Table 1: Comparison of 3DGS rendering and volume rendering methods.

| Method | Math Expression | Pros | Cons |
|---|---|---|---|
| 3DGS Rendering | Discrete summation | Fast rendering | High Mem consumption, Hard to fit 3D continuous surface |
| Volume Rendering | Continuous integration | Better 3D surface representation | Low rendering speed due to continuous sampling |

particularly in background areas. To overcome these challenges, the 2D Gaussian Splatting (2DGS) [14] proposes to use Gaussian surfaces as surfels to represent complex geometries [30], thereby improving the surface reconstruction quality. Particularly, 2DGS faces challenges when processing large-scale scenes, as it requires the explicit representation of a large number of Gaussian primitives, leading to significant GPU memory consumption. Therefore, 2DGS still exhibits limitations in novel view synthesis capabilities and the geometric representation of large-scale scenes.

In Table 1, we summarize the comparison of 3DGS rendering and volume rendering. To fully leverage the fast rendering advantages of Gaussian alpha blending while achieving effective implicit scene representation, we propose a novel Gaussian Voxel Kernel Functions (GVKF) method. Firstly, GVKF utilizes voxelization to implicitly represent 3DGS, managing the growth and pruning of Gaussian splats. This approach retains the expressive power of explicit Gaussian splats while enabling efficient management of these splats. Secondly, we carefully analyze the intrinsic connection between Gaussian splatting alpha blending rendering and traditional volume rendering from a mathematical perspective. We establish a 3DGS-based method to represent continuous scene opacity density fields through kernel regression. This makes it possible for discrete Gaussians to represent continuous scenes. By replacing the discrete opacity values in original 3DGS rendering pipeline (which can be viewed as collapsed kernel functions) with Gaussian kernel functions, we maintain the advantages of the original 3DGS alpha blending while optimizing the representation of continuous scenes. Moreover, we demonstrate that our proposed rendering method is mathematically consistent with traditional volume rendering. Thirdly, based on our constructed scene opacity representation, which is also known as the scene opacity field, we derive the bidirectional mapping relationship between opacity and the scene surface. This enables direct mesh extraction for scene surface. In summary, our contributions are as follows:

- We propose GVKF, an implicit continuous scene reconstruction method that integrates the effectiveness of implicit representation with the fast rasterization advantages of Gaussian Splatting, without the need for computationally intensive volume rendering.

- Based on GVKF, we further propose implicit representation of the scene surface, achieving efficient and high-quality scene surface reconstruction.

- Experiments demonstrate the usefulness of GVKF in open scenes, showcasing high-quality surface reconstruction accuracy, real-time rendering speeds, and significant savings in storage and memory consumption.

## 2 Related Works

### 2.1 Novel View Synthesis

The introduction of Neural Radiance Fields (NeRF) [27] has significantly advanced the development of 3D reconstruction and novel view synthesis. NeRF employs volumetric rendering techniques to intricately simulate the geometric structure of scenes and viewpoint-dependent characteristics, thereby considerably enhancing the quality of image rendering. Following NeRF, variants such as Mip-NeRF [1] and Zip-NeRF [2] have addressed the aliasing issues during rendering. Additionally, UC-NeRF [6], designed for outdoor scenes, enhances image consistency through color correction and pose refinement. InstantNGP [28] accelerates training and improves rendering efficiency by optimizing subvolume processing with grid pyramid techniques. Meanwhile, other feature grid-based scene representation methods [3, 22, 34, 5, 45] have been extensively explored to enhance the training capability and expressiveness of models. Recently, 3D Gaussian Splatting (3DGS) [15] effectively represents complex scenes using 3D Gaussian points, significantly boosting the efficiency of real-time high-resolution image rendering while maintaining rendering quality. Further research efforts like Scaffold-GS and Octree-GS [25, 32] have attempted more effective methods to organize and manage Gaussian points, which helps reduce memory usage and speed up training.

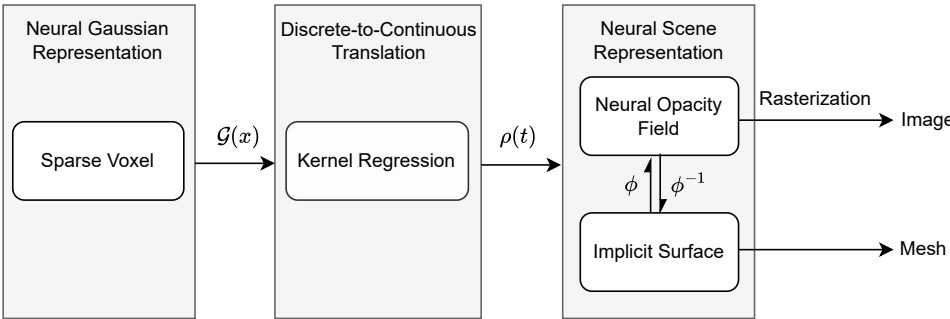

Figure 1: Framework of Gaussian Voxel Kernel Functions (GVKF) for scene representation. In this framework, discrete Gaussian primitives $\mathcal{G}$ represent continuous opacity density $\rho(t)$ on the ray via kernel regression. After slightly modifying the rasterization pipeline, the kernel function can be integrated into alpha blending rasterization without introducing dense points sampling. Additionally, we directly define the mapping relationship between the neural opacity field and the implicit surface.

## 2.2 Surface Reconstruction

Traditional isosurface extraction, relying on density thresholds, often struggles with fine details due to resolution and noise constraints. Recent studies propose more complex representation methods [48]. For instance, NeuS [37] uses MLP networks for occupancy grids or SDF, improving reconstruction accuracy and noise reduction [29, 43, 20, 7, 42]. Techniques like BakedSDF [44] translate the optimization of NeRF or neural SDFs into 3D meshes, enhancing features through high-resolution grids but increasing computational load. NeuS2 [39] introduces a novel formula for second-order derivatives with multi-resolution hash encoding and CUDA-based MLP technology, significantly reducing training time. StreetSurf [11] optimizes SDF mappings in open scenes and decouples static and dynamic objects. Despite advancements, NeRF-based methods still need optimization for processing speed and real-time rendering.

3DGS has gained attention for its high-quality scene reconstruction and rapid processing capabilities [15]. 3DGS uses multiple 3D Gaussian distributions with anisotropic covariance for precise control over scene attributes [49, 18]. This technology enhances surface reconstruction methods like SuGaR [12], which employs Poisson surface reconstruction for fast and accurate mesh extraction. However, irregular Gaussian sphere distribution affects surface quality. To improve this, 2DGS [14] uses 2D Gaussian planes for better surface conformity and TSDF for accurate reconstruction, though it may cause surface fragmentation. GOF [47] directly extracts surfaces using opacity thresholds and tetrahedral mesh extraction but is limited by high VRAM requirements. GSDF [46] combines 3DGS with a NeuS-like SDF branch for optimized rendering and reconstruction, increasing training time. Despite their potential, 3DGS-based methods face challenges like managing Gaussian points, high VRAM consumption, and degraded rendering quality.

## 3 Methods

As shown in Fig. 1, we first introduce the implicit neural 3DGS primitives representation based on a sparse voxel grid, which offers highly efficient storage management and the fitting power of neural networks. Secondly, we present our GVKF-based continuous scene representation, to explain its rationale, we have analyzed its intrinsic connections with Gaussian alpha blending [15] and traditional volume rendering [27] from a statistical analysis perspective. Finally, we describe the relationship between the proposed continuous scene representation (a neural opacity field) and implicit surface, and derive an explicit mapping function for mesh reconstruction.

### 3.1 Voxel Gaussian Representation

To achieve orderly 3DGS management while minimizing the explicit expression of them to save training storage consumption, we use a spatial sparse voxel grid to manage Gaussian primitives. During the initialization phase, the sparse grid is generated from the downsampled SfM point clouds

and dynamically grows or being eliminated during training. Each sparse grid is allowed to generate up to $m$ Gaussian primitives, and all these primitives are limited to a small range of space centered at the voxel grid.

**Gaussian Generation** For a particular 3D Gaussian expression, five attributes are required: $p \in \mathbb{R}^3$ (position), $\alpha \in \mathbb{R}$ (opacity), $R \in \mathbb{R}^{3 \times 3}$ (rotation matrix), $s \in \mathbb{R}^3$ (scaling), and $c \in \mathbb{R}^3$ (color). Then, a Gaussian $\mathcal{G}(x)$ can be generated as:

$$\mathcal{G}(x) = \alpha \cdot e^{-\frac{1}{2}(x-p)^T \sum^{-1}(x-p)}, \tag{1}$$

where $\sum \in \mathbb{R}^{3 \times 3}$ is covariance matrix defined as $\sum = Rss^T R^T$. $c$ is calculated via SH coefficients and camera direction. Different from traditional 3DGS[15] that treats them as explicit optimizable tensors, we decode them from a feature vector $\mathcal{F} \in \mathbb{R}^d$ via several MLPs:

$$\alpha = \text{MLP}_\alpha(\mathcal{F}, \text{camera}), R = \text{MLP}_R(\mathcal{F}), s = \text{MLP}_s(\mathcal{F}), c = \text{MLP}_c(\mathcal{F}, \text{camera}). \tag{2}$$

For alpha and color MLPs, the view camera and feature vector $\mathcal{F}$ are inputs, facilitating view-dependent fitting. Relative coordinates of Gaussians to the parent voxel center are stored with $\mathcal{F}$, compressing explicit Gaussian components and leveraging the MLP's fitting capacity. Gaussians are dynamically generated each iteration and recycled post-update, reducing memory usage.

**Voxel Registration**. To control Gaussian numbers in large open scene, we eschew the traditional adaptive density control strategy, adopting a method inspired by scaffold-Gaussian [25] and Octree-Gaussian [32]. The voxel registration is based on gradient accumulation. After each iteration, gradients from 3DGS are recorded and accumulated in their respective voxels, denoted as $\nabla$. Voxels where $\nabla$ exceeds a set threshold are subdivided into eight subvoxels to increase grid resolution, continuing until the maximum depth is reached. Additionally, less frequently used voxels are discarded after a specified period.

## 3.2 Neural Opacity Field of 3DGS

Since 3DGS rasterization rendering and traditional volume rendering share some overlapping concepts, in this section, we sort them out and introduce our method from a statistical perspective while avoiding introducing redundant mathematical symbols.

**Continuous Scene Description**. We define $\rho(t) : [0, +\infty] \rightarrow [0, 1]$ as the opacity density function, which measures the probability of a ray encountering a particle at position $t$. We define $\mathcal{T}(t) : [0, +\infty] \rightarrow [0, 1]$ as the transmission function, which measures the probability that a ray has not encountered any particles from its origin to point $t$. Considering the probability that a ray does not encounter any particles at time step $t + dt$, denoted as $\mathcal{T}(t + dt)$, it is evident that $\mathcal{T}(t + dt) = \mathcal{T}(t)(1 - \rho(t)dt)$. Solving this differential equation, we obtain the relationship between $\mathcal{T}(t)$ and $\rho(t)$:

$$\mathcal{T}(t) = \exp(-\int_0^t \rho(t)dt). \tag{3}$$

Therefore, we obtain the cumulative distribution function (CDF) of the probability that a ray **hits** a particle over the interval $[0, t]$: $\Phi(t) = 1 - \mathcal{T}(t)$, with the corresponding probability density function (PDF) being $\Phi'(t) = \mathcal{T}(t) \cdot \rho(t)$. From the perspective of volume rendering, this PDF is used as the probability of the appearance of color along the ray, ultimately taking the mathematical expectation of the color as the ray color:

$$C = \int_0^B \mathcal{T}(t) \cdot \rho(t) \cdot c(t)dt + \mathcal{T}(B) \cdot c_{bg}. \tag{4}$$

The discrete formulation of volume rendering Eq. 4 is:

$$C = \sum_{i=1}^N T_i \cdot \alpha_i \cdot c_i, \quad \alpha_i = (1 - \exp(-\sigma_i \delta_i)), \quad T_i = \prod_{j=1}^{i-1}(1 - \alpha_j) \tag{5}$$

where opacity $\alpha_i$ represents the accumulated result in a sampling interval $\delta_i$ of volume density $\sigma_i$, hence the value of $N$ does not influence the result as long as $\alpha_i$ is adapted enough. Based on the similar idea of volume rendering, the PDF $\Phi'(t) = \mathcal{T}(t) \cdot \rho(t)$ can also be reasonably considered

as the probability of the appearance of a surface along the ray, where the place with the highest probability density is most likely to have a surface. Correspondingly, on the CDF $\Phi(t)$, this is the place where the derivative is the largest. In this paper, we use the CDF $\Phi(t)$ to describe continuous scenes based on the camera rays, to facilitate integration with the 3DGS rasterization rendering pipeline. In section 3.3 , we will prove that under the 3DGS representation, the place where the derivative of CDF is the largest is not actually the surface, so a method to locate the surface will be introduced.

**Kernel Regression of 3DGS**.

In implicit scene representation methods based on volume rendering, the continuous opacity density function is directly predicted by a MLP[27]. In our approach, the continuous opacity density function $\rho(t)$ is fitted through kernel regression via discrete Gaussian primitives after a differentiable transformation: $\mathcal{G}_i(x, y, z) \rightarrow \mathcal{K}_i(t - t_i)$ from 3D to 1D. The transformation consists of three steps: (1). The Gaussian primitives that the ray passes through are selected as kernel functions. (2). According to the Ray-Gaussian Intersection method [16], the ray is transformed into the local coordinate system of each 3DGS to obtain the 1D probability density alone the ray. Here, the peak of the 1D probability density, denoted as $t_i$, is defined as the Ray-Gaussian Intersection [10, 16, 47], indicating that the 3DGS has the greatest influence at this point alone the ray. (3). To integrate with regularization methods [14, 47], we assume that each 3DGS fits the surface of the object. There-

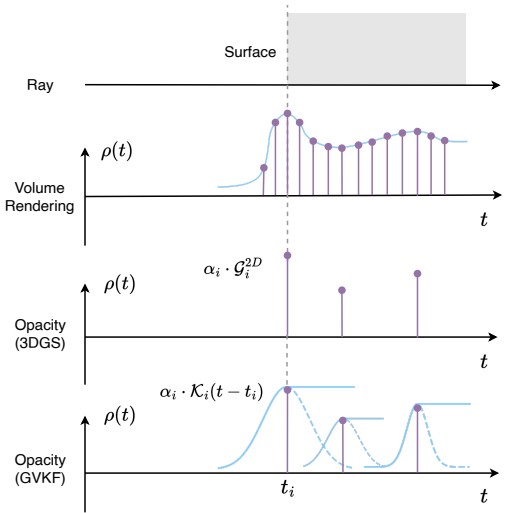

Figure 2: Comparison of Volume Rendering, 3D Gaussian Splatting with Alpha Blending, and GVKF Rendering.

fore, after $t_i$, the probability density continues to remain at its maximum value, indicating that the object is solid. Without loss of generality, the opacity density $\rho(t)$ on camera ray can be expressed as:

$$\rho(t) = \sum_i^N \alpha_i \cdot \mathcal{K}_i(t - t_i), \quad \mathcal{K}_i(t) = \begin{cases} \exp(-k_i \cdot t^2) & t < 0 \\ 1 & t \geq 0, \end{cases} \tag{6}$$

where $N$ represents the number of activated kernel functions along the ray, and $k_i$ represents the summarized transform of Ray-Gaussian transform (See Appendix A.1 for details). $\alpha_i$ represents the coefficient for each kernel function.[1]

**Rendering**. As for traditional 3DGS rasterization, the pixel color is rendered through alpha blending on $N$ 3DGS being passed through by the ray:

$$C = \sum_{i=1}^N c_i \cdot \alpha_i \cdot \mathcal{G}_i^{2D} \prod_{j=1}^{i-1}(1 - \alpha_j \cdot \mathcal{G}_j^{2D}) \tag{7}$$

In this scenario, $\alpha_i$ is constant value representing the opacity of Gaussians. This point-based rendering is coherent with Eq. 5, with extremely sparse sampling points to simulate dense volume rendering. However, it is impossible to recover continuous opacity density alone the ray from such a rendering equation, as illustrated in row-3 of Fig. 2. This is because the third row of the covariance matrix of 3DGS is discarded, and it is directly projected onto a 2D plane to evaluate the impact on the opacity of points along the ray. From the perspective of Eq. 6, this means that along the ray, the influence range of all $N$ kernel functions that intersect with the ray collapses to an infinitesimally value, making it impossible to recover a continuous opacity density field. To solve this, Eq. 7 can be modified to: [2]

$$C = \sum_{i=1}^N c_i \cdot \alpha_i \cdot \mathcal{K}_i(0) \prod_{j=1}^{i-1}(1 - \alpha_j \cdot \mathcal{K}_j(0)) \tag{8}$$

---

[1]Specifically, $\alpha_i = \beta_i \frac{\sqrt{k_i}}{\sqrt{\pi}}$, where $\beta_i$ is a constant value representing opacity related to Gaussian primitives.
[2]This rendering form is firstly proposed by GOF [47]

This equation will not affect the goal of 3DGS rendering: to approximate traditional volume rendering using sparse sampling points. And it allows for broadening the collapsed kernel functions to fit the continuous opacity function of the scene.

**Scene Representation**. Based on the discussion at the beginning, the scene surface can be described via CDF $\Phi(t)$ on the ray in a continuous way, which can be calculated like Eq. 8 (removing color):

$$\Phi(t) = \sum_{i=1}^{N} \alpha_i \cdot \mathcal{K}_i(t - t_i) \prod_{j=1}^{i-1}(1 - \alpha_j \cdot \mathcal{K}_j(t - t_j)) \tag{9}$$

### 3.3 Implicit Surface Mapping

This implicit opacity field (denoted as neural opacity field since it is represented by neural Gaussians) measures the CDF of the probability that a ray hits solid scene surface. In the next section, we introduce the mapping of $\Phi(t)$ to implicit surface.

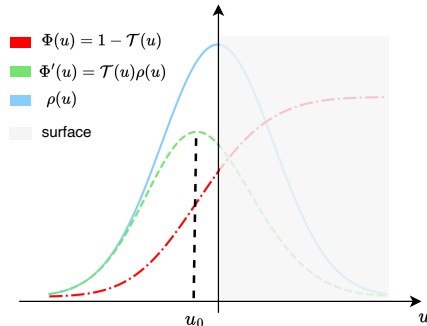

We represent implicit surface with signed distance function (SDF), denoted as function $D(t)$ on the camera ray. To recover $D(t)$ of given $\Phi(t)$ that is calculated from well trained 3DGS, we firstly study the reverse mapping problem: $\phi : D(t) \to \Phi(t)$

Figure 3: Illustration of functions $\Phi(u), \Phi'(u), \rho(u)$.

**Opacity Density Near the Surface**. To ensure that the 3DGS aligns with the object's surface and thus reflects the object's shape, depth distortion regularization [14, 47] is introduced during the Gaussian training process. This encourages the distribution of 3DGS along the ray to aggregate together, causing the peak of the kernel functions to coincide with the object's surface. In the next discussion, the coordinate of object surface on the ray is assumed as $t^*$ with $D(t^*) = 0$. Considering $\rho(t)$ at the interval $t \in [0, t^*]$, we have:

$$\rho(t) = \rho(t^* - D(t)) = \sum_{j=1}^{M} \alpha_j \cdot \exp(-k_j \cdot D(t)^2), \quad 0 \leq t \leq t^* \tag{10}$$

Where $M$ represents the number of Gaussian kernels concentrated on the surface. To facilitate calculations, we convert the opacity density system to the SDF coordinate system, with $t^*$ as the origin and letting $u = -D(t)$, as illustrated in Fig. 3, we have:

$$\Phi'(u) = \mathcal{T}(u) \cdot \rho(u) = \exp(-\int_{-t^*}^{u} \rho(w)dw) \cdot \rho(u) \tag{11}$$

$$\Phi''(u) = -\rho^2(u) \cdot \exp(-\int_{-t^*}^{u} \rho(w)dw) + \rho'(u) \cdot \exp(-\int_{-t^*}^{u} \rho(w)dw)$$

$$= [-\rho^2(u) + \rho'(u)] \exp(-\int_{-t^*}^{u} \rho(w)dw) \tag{12}$$

where $\rho(u) \sim \mathcal{N}(0, \sigma^2), \sigma^2 = \sum_{i=1}^{M} \frac{1}{2\pi\alpha_i^2}$, which can be directly derived from the additive property of the normal distribution. Then letting $h(u) = -\rho^2(u) + \rho'(u)$, we have:

$$h(u) = -\rho^2(u) + \rho'(u)$$

$$= -\rho(u)[\rho(u) + \frac{u}{\sigma^2}] \tag{13}$$

It is easy to prove that $h(u)$ crosses a unique zero point $u_0$ from top to bottom on the u-axis, and $u_0 < 0$. This means that the peak of $\Phi'(u)$ will appear before the surface, so it is not reasonable to simply determine the actual intersection point of the light ray with the surface by directly evaluating the peak of $\Phi'(u)$. To locate the accurate surface, a transcendental equation of $u$ is needed to be

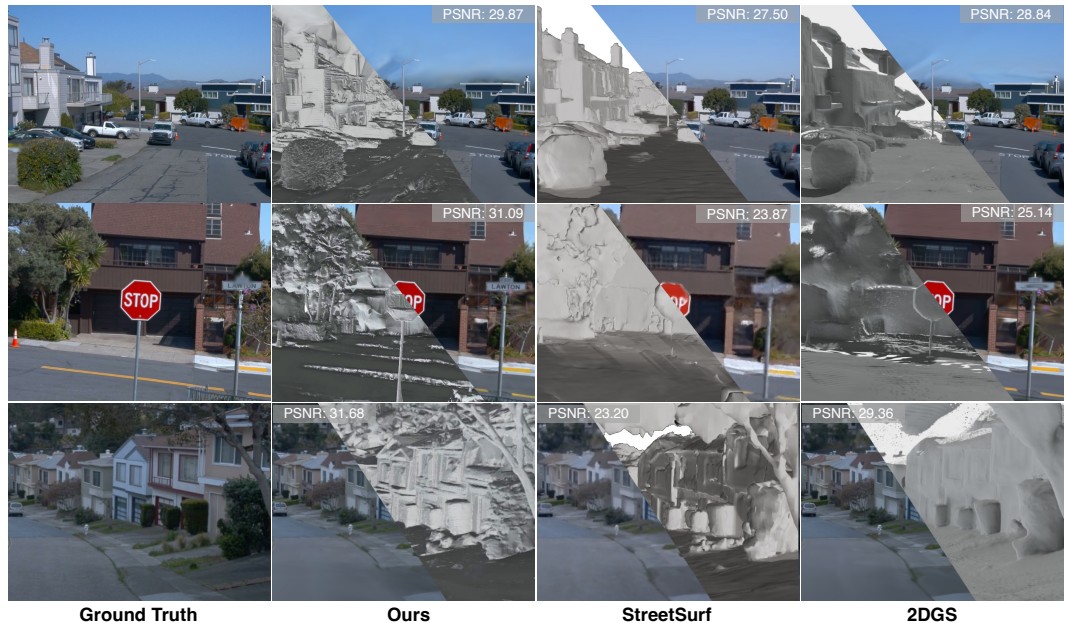

| Ground Truth | Ours | StreetSurf | 2DGS |
|---|---|---|---|

Figure 5: Qualitative comparison of novel view synthesis and surface reconstruction on the Waymo Open Dataset [35], with each subplot annotated with PSNR values to quantify image quality. Our method shows higher geometric precision and detail, validating its efficiency and superiority in processing open scenes, especially in geometric accuracy and detail reproduction.

solved to get $u_0$:

$$\rho(u) = -\frac{u}{\sigma^2}, \quad \rho(u) = \frac{1}{\sqrt{2\pi}\sigma}\exp(-\frac{u^2}{2\sigma^2}), \quad \sigma^2 = \sum_{i=1}^{M}\frac{1}{2\pi\alpha_i^2} \tag{14}$$

It is impossible to directly get the analytical solution, however, numerical computation methods can be applied to solve $u_0$. This may require some extra time for computation.

**Mapping from Opacity to Surface**.

Based on the analysis above, we can always have exact number of $u_0$ via numerical computation method. However, it is hard to find out the inverse function of $\Phi(u)$ for directly building the mapping of $\Phi(t)$ to $D(t)$. For the balance of surface smoothing while reducing the indelible error, we represent mapping relationship of $u \to \Phi(u)$ via Logistic Function as follows:

$$\Phi(u) = \frac{1}{1 + \exp(-\mu(u - u_0))} \tag{15}$$

Figure 4: Illustration of opacity to SDF mapping.(Eq. 16)

where $\mu$ represents the smooth factor. We choose Logistic Function because of its formal is concise and shares similar shape of $\Phi(u)$. More importantly, it only has one inflection point at $(0, 0.5)$, which can be used to simulate the inflection point of $\Phi(u)$ after translation. Finally, we represent implicit SDF function via Inverse function transformation of Logistic Function, as shown in Fig. 4:

$$D(t) = \ln(\frac{1}{\Phi(t)} - 1)/\mu - u_0 \tag{16}$$

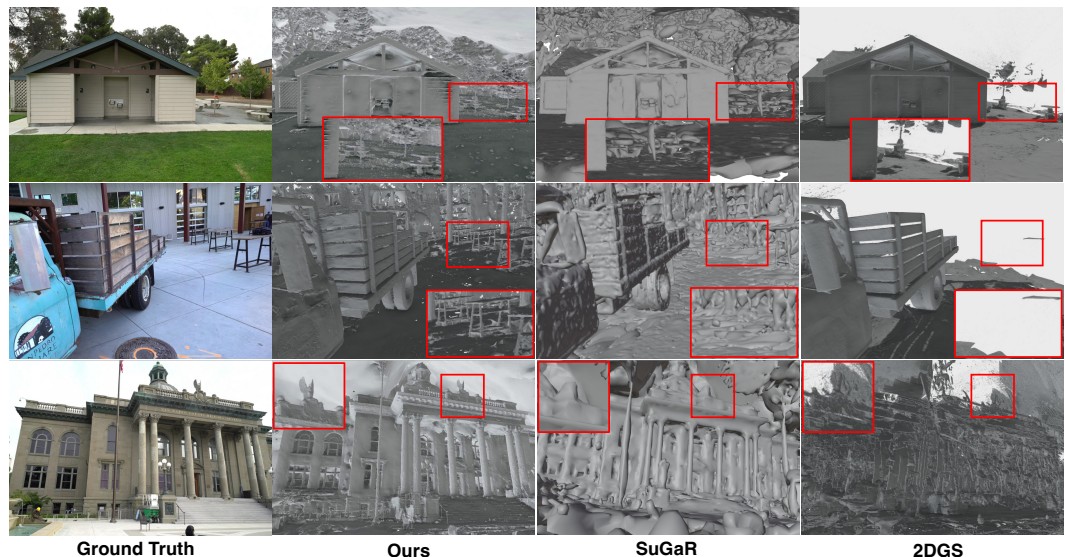

| Ground Truth | Ours | SuGaR | 2DGS |
|---|---|---|---|

Figure 6: Qualitative comparison on the Tanks and Temples dataset [17] shows that our method excels in reconstructing complex backgrounds with high geometric granularity. In contrast, 2DGS often results in fragmented backgrounds, while SuGaR displays uneven spherical shapes, affecting both visual and geometric quality.

Table 2: Quantitative evaluation of novel view synthesis and surface reconstruction on the Waymo Open Scene dataset [35]. Using LiDAR data as ground truth, we calculated Chamfer Distance (C-D) values for reconstruction accuracy. Our method performs excellently in both novel view synthesis and surface reconstruction, outperforming other methods in Gaussian point usage, VRAM occupancy, and real-time rendering.

| Method | PSNR $\uparrow$ | C-D $\downarrow$ | MB (Storage) $\downarrow$ | GB (GPU) $\downarrow$ | FPS $\uparrow$ | Training Time $\downarrow$ |
|---|---|---|---|---|---|---|
| NeuS | 13.24 | **0.76** | 170 | 31 | $\sim 0.1$ | 5 h |
| F$^2$-NeRF | 24.70 | 886.77 | 130 | 24 | $\sim 0.1$ | 0.8 h |
| StreetSurf | 27.12 | 1.02 | 540 | 22 | $\sim 0.1$ | 1.5 h |
| 3DGS | 27.99 | 3.57 | 230 | 23 | **63** | 0.75 h |
| SuGaR | 23.71 | 3.08 | 228 | 33 | 56 | 1.5 h |
| 2DGS | 28.51 | 1.67 | 238 | 23 | 51 | **0.7 h** |
| GVKF (Ours) | **30.24** | 1.57 | **30** | **14** | 32 | 1.5 h |

## 4 Experiments

### 4.1 Experimental Settings

**Datasets.** To assess our method's performance against baseline methods in open scenes, we used three datasets. We first experimented with the Waymo Open Scene dataset [35], using three cameras per scene from five available, each scene containing about 600 images. We employed LiDAR point clouds to evaluate reconstruction quality, although LiDAR data was not used as training input. We also tested on the Tank and Temple dataset [17], which includes trajectories and ground truth for six selected scenes. Lastly, we evaluated the Mip-NeRF 360 dataset [1]; due to the absence of ground truth, our focus was on novel view synthesis to demonstrate our method's efficacy in this aspect.

**Baselines.** In terms of surface reconstruction, we presented the results on the Waymo dataset [35] in tables 2 and figures 5, comparing state-of-the-art implicit methods (such as NeuS [37], F$^2$-NeRF [38], StreetSurf [11]) and explicit methods (such as 3DGS [15], SuGaR [12], 2DGS [14]). We utilized PSNR to evaluate the results of novel view synthesis and Chamfer distance to measure reconstruction accuracy, while also recording training time, VRAM usage, and the size of the Gaussian point files post-training. Additionally, as shown in Table 3 and Figure 6, we conducted comparisons on the Tank

Table 3: Quantitative evaluation on the Tanks and Temples dataset [17] using F1 scores and training time as metrics. Our method outperforms all existing explicit methods in F1 scores and is comparable to implicit methods in reconstruction accuracy, with significantly reduced training time. These results highlight our method's efficiency and accuracy. Comparison of concurrent work GOF [47] is presented in Appendix A.4.

| Method | Implicit | | | Explicit | | | |
|---|---|---|---|---|---|---|---|
| | NeuS | Geo-NeuS | Neuralangelo | SuGaR | 3DGS | 2DGS | Ours |
| Barn | 0.29 | 0.33 | **0.70** | 0.14 | 0.13 | 0.36 | **0.40** |
| Caterpillar | 0.29 | 0.26 | **0.36** | 0.16 | 0.08 | 0.23 | **0.34** |
| Courthouse | 0.17 | 0.12 | **0.28** | 0.08 | 0.09 | 0.13 | **0.25** |
| Ignatius | 0.83 | 0.72 | **0.89** | 0.33 | 0.04 | 0.44 | **0.51** |
| Meetingroom | 0.24 | 0.20 | **0.32** | 0.15 | 0.01 | 0.16 | **0.23** |
| Truck | 0.45 | 0.45 | **0.48** | 0.26 | 0.19 | 0.26 | **0.40** |
| Mean | 0.38 | 0.35 | **0.50** | 0.19 | 0.09 | 0.30 | **0.36** |
| Time | >24 h | >24 h | >24 h | >1 h | ~**15 min** | ~30 min | ~1.5 h |

and Temple dataset with implicit methods (such as NeuS [37], Geo-NeuS [9], Neuralangelo [20]) and explicit methods (such as 3DGS [15], SuGaR [12], 2DGS [14]). We used official scripts to evaluate F1 scores. For novel view synthesis, we compared various advanced methods on the Mip-NeRF 360 dataset, including NeRF [27], Deep Blending [13], Instant NGP [28], MERF [31], Mip-NeRF 360 [1], BakedSDF [44], 3DGS [15], SuGaR [12], and 2DGS [14]. We use evaluation metrics such as PSNR, SSIM, and LPIPS.

**Implementation Details** Our method modifies the representation of 3DGS and slightly adjusts the opacity weights in the rendering pipeline using Gaussian kernel functions. This ensures compatibility with other components of Gaussian rasterization rendering. Similarly, we employ the same L1 loss and D-SSIM loss as 3DGS to supervise color loss, and we use the same Gaussian regularization term as 2DGS and GOF to promote alignment between the Gaussians and the surface. After training, the SDF field of the scene can be directly extracted based on Eq. 16 and exported to a mesh with the MC[24]/MT[8, 33] algorithm. To export complete sky and background, the modified MT algorithm in GOF[47] is used.

Table 4: Quantitative evaluation on the Mip-NeRF 360 [1] outdoor scene dataset is presented. Since the dataset lacks ground truth for surface reconstruction, we assessed the results of novel view synthesis.

| Method | PSNR $\uparrow$ | SSIM $\uparrow$ | LPIPS $\downarrow$ |
|---|---|---|---|
| NeRF | 21.46 | 0.458 | 0.515 |
| Deep Blending | 21.54 | 0.524 | 0.364 |
| Instant NGP | 22.90 | 0.566 | 0.371 |
| MERF | 23.19 | 0.616 | 0.343 |
| Mip-NeRF 360 | 24.47 | 0.691 | 0.283 |
| BakedSDF | 22.47 | 0.585 | 0.349 |
| 3DGS | 24.24 | 0.705 | 0.283 |
| SuGaR | 22.76 | 0.631 | 0.349 |
| 2DGS | 24.33 | 0.709 | 0.284 |
| GVKF (Ours) | **25.47** | **0.757** | **0.240** |

## 4.2 Analysis

Figure 5 demonstrates the superiority of our method in capturing detailed features of roadside houses, bushes, and other objects. In contrast, the 2DGS method produced more holes and fragmentation, while the StreetSurf method lost some critical geometric features. The results in Table 2 indicate that our method surpasses other methods in terms of view synthesis and reconstruction accuracy, and it requires fewer Gaussian points and VRAM for large-scale scene reconstructions. Figure 6 highlights our method's excellent performance in scene restoration. The SuGaR method generated excessive irregular protrusions, and 2DGS exhibited more fragmentation and floating debris. According to the results in Table 3, our method outperforms all explicit methods and achieves comparable reconstruction results to implicit methods, while maintaining equivalent GPU time usage. The results in Table 4 confirm that our method leads in novel view synthesis across all compared methodologies.

## 4.3 Ablation Study

In this section, we evaluate the impact of varying voxel grid sizes on the neural Gaussians by conducting experiments with the Waymo datasets. We selected voxel sizes of $1, 0.1, 0.01, 0.001$, as

Table 6: Further ablation study on voxel Gaussian representation and SDF mapping. w/o voxel: We eliminate the using of MLPs and voxel grid, w/o sdf: we directly use linear assumption between opacity function and SDF function.

| Ablation | PSNR | F1 | Mem (GB) | Storage (MB) | Training Time | Meshing Time |
|---|---|---|---|---|---|---|
| Ours | 26.31 | 0.36 | $\sim 9$ G | 90 M | $\sim 1.5$ h | $\sim 15$ min |
| w/o voxel | 23.60 (-2.71) | 0.39 (+0.03) | $\sim 16$ G ($\times 1.6$) | 467 M ($\times 5.2$) | $\sim 1.4$ h | $\sim 15$ min |
| w/o sdf | 26.31 | 0.30 (-0.06) | $\sim 9$ G | 90 M | $\sim 1.5$ h | $\sim 15$ min |

presented in Table 5. When the voxel size is too large, the sparse neural Gaussians fail to learn the scene representation and return NaN errors. As the number of voxels increases, more Gaussians are generated for scene representation, thereby enhancing the quality of novel view synthesis. However, the improvements plateau when the voxel size is reduced to 0.001, which also requires more training time and becomes impractical. Therefore, we set the voxel size to 0.01 to balance training time and rendering quality.

We further conducted ablation study on the Tanks and Temples dataset [17] to evaluate the impact of voxel representation and SDF mapping. The results are presented in Tab. 6. It can be observed that utilizing voxel representation significantly improves the PSNR for NVS tasks and reduces memory consumption dramatically

Table 5: Influence of different voxel size.

| Voxel Size | Initial voxels | Final voxels | PSNR | Time |
|---|---|---|---|---|
| 1 | $\sim 2$ k | - | - | - |
| 0.1 | $\sim 80$ k | $\sim 110$ k | 29.34 | 1.2 h |
| 0.01 | $\sim 90$ k | $\sim 1100$ k | 30.24 | 1.5 h |
| 0.001 | $\sim 100$ k | $\sim 1100$ k | 30.29 | 4 h |

compared to naive 3DGS setup. Although there is a slight decrease in the geometric quality of surface reconstruction, we consider this trade-off acceptable.

### 4.4 Limitation

Implicit methods, such as those based on NeRF [27, 37, 20], typically utilize a global fitting approach for SDF, which allows them to fully leverage the universal approximation capabilities of MLPs. This is advantageous even in areas with sparse viewpoints. However, our current method employs a local line-of-sight-based SDF fitting, a compromise made to adapt to the 3DGS rendering style. This means that regions not covered by the training viewpoints lack fitting capability, resulting in uneven surfaces.

In addition, While our method advances 3D surface reconstruction in open scenes, it faces challenges with dynamic objects and the decoupling of distant and near views, sometimes misrepresenting the sky as a surface enveloping the model. The lack of sufficient prior knowledge for optimizing complex scenes also poses limitations.

## 5  Conclusion

This paper introduces GVKF, combining Gaussian splatting's rapid rasterization with the efficiency of implicit expressions to enhance reconstruction quality and speed significantly. By employing a voxelized implicit representation of 3DGS, GVKF retains the expressive power of explicit Gaussian maps while managing them effectively. We have explored the relationship between Gaussian splatting's alpha blending and traditional volume rendering, developing a GS-based method to represent continuous scene opacity density fields through kernel regression, addressing 3DGS's limitations in continuous scene representation.

Experimental results demonstrate GVKF's effectiveness in open scenes, showing notable improvements in reconstruction accuracy, real-time rendering speeds, and reductions in storage and memory usage. These advancements support applications in fields like autonomous driving and virtual reality, pushing forward surface reconstruction technology.

## Acknowledgements

This research is supported by the National Natural Science Foundation of China (No. 62406267), Guangzhou-HKUST(GZ) Joint Funding Program (Grant No.2023A03J0008), Education Bureau of Guangzhou Municipality, and Guangzhou Quwan Network Technology Co., Ltd.

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

# A Appendix / Supplemental Material

## A.1 Ray-Gaussian Intersection

Based on Eq. 1 in the main text, the influence of 3DGS in camera space on a one-dimensional ray can be expressed as follows:

$$\rho(t) = \exp(-\frac{1}{2}(vt - p)\Sigma^{-1}(vt - p)) \qquad (17)$$

Here, $v$ represents the unit vector of the ray direction. This formula converts the three-dimensional influence of 3DGS into a one-dimensional function along a specific camera ray, which is a one-dimensional Gaussian function. Fig. 7 demonstrates the relationship of this transform. For ease of notation, we express it as:

$$\rho(t) = \exp(-k_i \cdot (t - t_i)^2) \qquad (18)$$

where $t_i$ denotes the point along the ray where 3DGS has the maximum impact, also known as the "ray-Gaussian intersection," which can be analytically given by:

$$t_i = \frac{p^T \Sigma^{-1} v}{v^T \Sigma^{-1} v} \qquad (19)$$

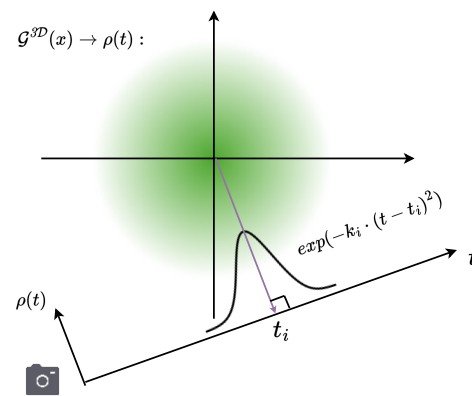

Figure 7: Ray-Gaussian Intersection in local 3DGS coordinate.

More proof details can be found in "Approximate Differentiable Rendering with Algebraic Surfaces." [16]

## A.2 More Implementation Details

As demonstrated in the ablation experiments, to balance quality and speed, we chose to downsample the initial Gaussian point cloud using a voxel size of 0.01. Within each voxel, the dimension of $\mathcal{F}$ is set to 32, and it stores the relative coordinates of 10 Gaussian points, indicating that the maximum number of Gaussians generated per voxel grid is 10. Gaussians with an opacity less than 0 will be hidden during each iteration. In each scene, all voxel grids share a total of four MLPs, which decode different Gaussian attributes from the corresponding voxels.

Regarding voxel registration, the gradient threshold is empirically set to $2 \times 10^{-4}$, meaning that voxel grids with an average gradient exceeding this value after each iteration will be subdivided using an octree method. The maximum recursion depth is set to 3 to control the number of Gaussians in the scene, ensuring it does not exceed a certain threshold. Voxel evaluation is performed every 500 iterations to determine which voxels should be subdivided or reclaimed. For other settings, we strive to remain consistent with the original 3DGS settings.

## A.3 More Results

Our method focuses on the challenging task of open scene reconstruction. Here, we provide a comprehensive quantitative comparison with other related methods on the Mip360 dataset, as shown in Table 7. Additionally, we have included more experimental results on the Mip360 and Tank and Temple datasets, as shown in Figures 8 and 9. For more qualitative results, please visit the project page.

**Discussion on indoor scene.** We observe that current methods based on 3DGS perform adequately for indoor scenes, where there is typically 360-degree viewpoint coverage. However, they underperform in outdoor scenes due to limited viewpoint coverage. Heuristic splitting and pruning strategies in original 3DGS tend to fit the training viewpoints rather than distributing evenly across the space. This leads to poorer novel view synthesis results in outdoor environments. As illustrated in Fig. 10, without a voxel grid, heuristic Gaussian growth strategies result in uneven spatial distribution of GS, sometimes even creating holes. Conversely, using voxel grids to constrain Gaussians allows for efficient management of their spatial distribution, supporting better novel view synthesis.

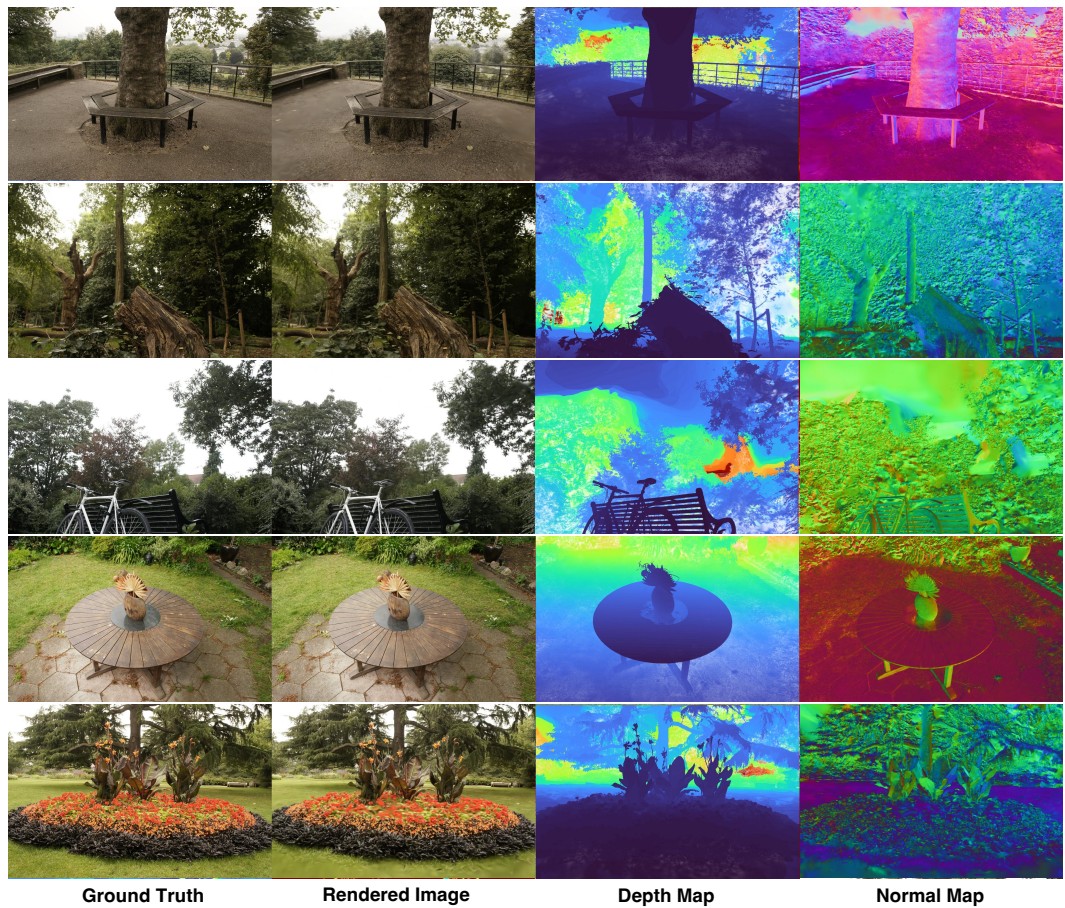

|  Ground Truth | Rendered Image | Depth Map | Normal Map |

Figure 8: Additional experimental results on the Mip-NeRF360 dataset [1]. From left to right: Ground Truth, Novel View Synthesis, Rendered Depth Map, and Normal Map.

Table 7: Quantitative evaluation on the Mip-NeRF 360 [1]. All scene dataset is presented.

| Method | Outdoor Scene | | | Indoor Scene | | | Average | | |
|---|---|---|---|---|---|---|---|---|---|
| | PSNR ↑ | SSIM ↑ | LPIPS ↓ | PSNR ↑ | SSIM ↑ | LPIPS ↓ | PSNR ↑ | SSIM ↑ | LPIPS ↓ |
| NeRF | 21.46 | 0.458 | 0.515 | 26.84 | 0.79 | 0.37 | 23.85 | 0.61 | 0.45 |
| Deep Blending | 21.54 | 0.524 | 0.364 | 26.4 | 0.844 | 0.261 | 23.70 | 0.67 | 0.32 |
| Instant NGP | 22.9 | 0.566 | 0.371 | 29.15 | 0.88 | 0.216 | 25.68 | 0.72 | 0.30 |
| MERF | 23.19 | 0.616 | 0.343 | 27.8 | 0.855 | 0.271 | 25.24 | 0.72 | 0.31 |
| MipNeRF360 | 24.47 | 0.691 | 0.283 | **31.72** | 0.917 | **0.18** | 27.69 | 0.79 | 0.24 |
| BakedSDF | 22.47 | 0.585 | 0.349 | 27.06 | 0.836 | 0.258 | 24.51 | 0.70 | 0.30 |
| Mobile-NeRF | 21.95 | 0.470 | 0.470 | 12.19 | 0.26 | 0.26 | 17.07 | 0.36 | 0.37 |
| 3DGS | 24.24 | 0.705 | 0.283 | 30.99 | **0.926** | 0.199 | 27.24 | 0.80 | 0.25 |
| SuGaR | 22.76 | 0.631 | 0.349 | 29.44 | 0.911 | 0.216 | 25.73 | 0.76 | 0.29 |
| 2DGS | 24.33 | 0.709 | 0.284 | 30.39 | 0.924 | 0.182 | 27.02 | 0.80 | 0.24 |
| GVKF (Ours) | **25.47** | **0.757** | **0.240** | 30 | 0.915 | 0.2 | **27.48** | **0.83** | **0.22** |

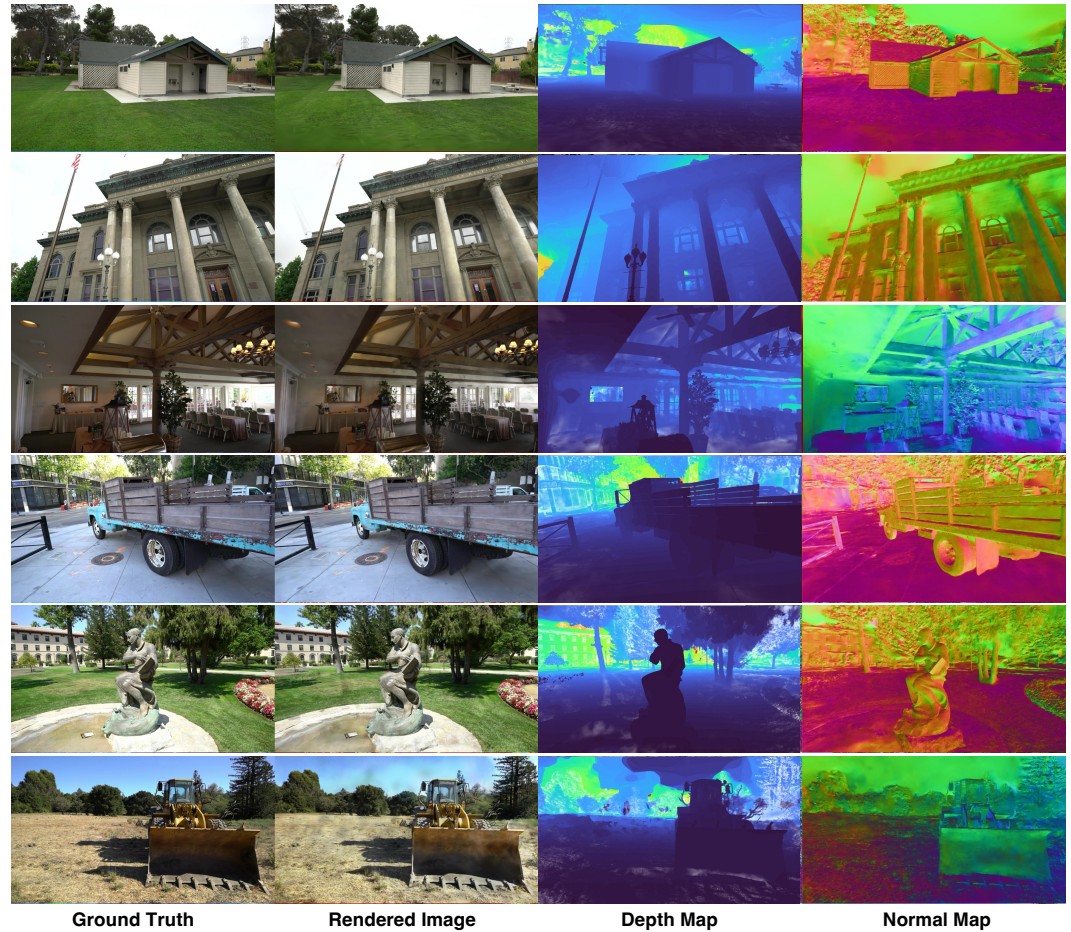

| Ground Truth | Rendered Image | Depth Map | Normal Map |

Figure 9: Additional experimental results on the Tanks and Temples dataset [17]. From left to right: Ground Truth, Novel View Synthesis, Rendered Depth Map, and Normal Map.

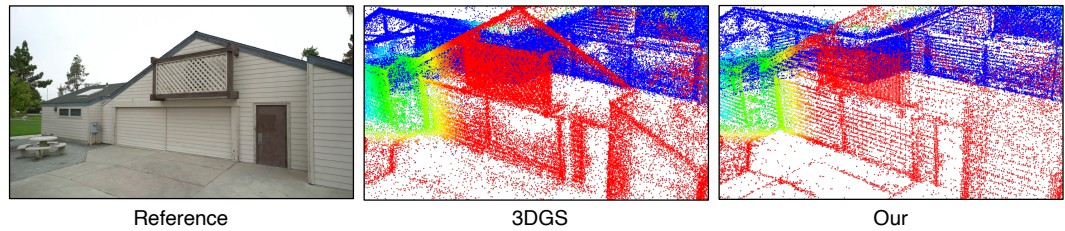

| Reference | 3DGS | Our |

Figure 10: GVKF Gaussian point visualization compared to traditional Gaussian method.

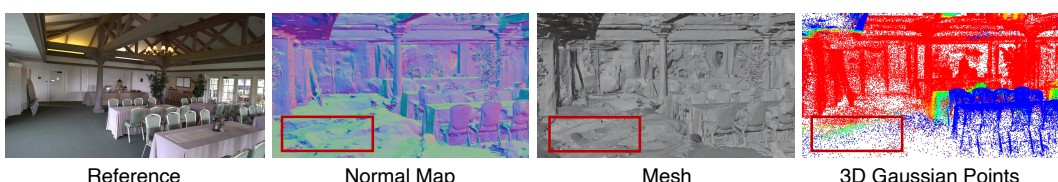

| Reference | Normal Map | Mesh | 3D Gaussian Points |

Figure 11: Failure case. The sparse view area with less Gaussians tends to appear uneven surface.

Table 8: NVS and storage comparation to GOF on Mip-NeRF 360 Dataset [1].

| Mip-NeRF 360 | PSNR ↑ | SSIM ↑ | LPIPS ↓ | Storage ↓ |
|---|---|---|---|---|
| GOF | 24.53 | 0.733 | 0.245 | 649 M |
| GVKF (ours) | **25.47** | **0.757** | **0.240** | **68 M** |

Table 9: Mesh quality comparison (F1-score) to GOF on Tanks and Temples Dataset. [17]

| TanT | Barn | Caterpillar | Courthouse | Ignatius | Meetingroom | Truck | Mean |
|---|---|---|---|---|---|---|---|
| GOF | **0.51** | **0.41** | **0.28** | **0.68** | **0.28** | **0.59** | **0.46** |
| GVKF (ours) | 0.40 | 0.34 | 0.25 | 0.51 | 0.23 | 0.40 | 0.36 |

**Failure Case.** As shown in Fig. 11, in areas with sparse viewpoint coverage, the distribution of 3DGS is sparse and irregular, which hinders the fitting of smooth planes. This sparsity compromises the integrity of the surface reconstruction, resulting in models that are geometrically inaccurate. Our method still struggles to effectively address these issues.

### A.4   Comparation to Gaussian Opacity Field

The similar rendering equation is firstly proposed by GOF [47], while this work provides in-depth analysis of the relationship among this rendering strategy, volume rendering and Gaussian alpha blending. Different from GOF, our scene representation is implicit, addressing the common issue of high memory consumption faced by 3D Gaussian splatting. Additionally, we developed a mapping function from opacity to SDF to alleviate the influence of directly linear transform between these fields.

As shown in Tab. 8, GOF uses explicit Gaussian management, still faces high storage consumption issues, making training large scenes on a single card challenging. Our method achieves better novel view synthesis results with less storage usage. However, as shown in Tab. 9, our current implementation has some geometric precision gaps compared to GOF, the potential reasons may include:

- GOF's iterative optimization extraction method achieves more precise isosurfaces than ours. (May require long meshing time $\sim 2$ h)
- Further adaptation of regularization term to voxel grids might be needed in our method to improve mesh quality.

