# OpenReview forum: "GVKF: Gaussian Voxel Kernel Functions for Highly Efficient Surface Reconstruction in Open Scenes"
_NeurIPS.cc/2024/Conference — NeurIPS 2024 poster_

### Official Review · Reviewer_6M18 · 2024-06-26

**Soundness:** 3
**Presentation:** 2
**Contribution:** 3
**Rating:** 5
**Confidence:** 2

**Summary:**

This paper presents Gaussian Voxel Kernel Functions (GVKF) for 3D surface reconstruction. The authors establish a continuous signed distance function derived from discrete 3D Gaussians, achieving high-fidelity open-scene surface reconstruction. They claim that the proposed method has high reconstruction quality, real-time rendering speeds, and significant memory efficiency in both runtime and training time. However, these claims lack sufficient discussion or validation throughout the paper.

**Strengths:**

1. The voxel kernel function is a relatively new concept and appears interesting.

2. The evaluation is extensive. The paper conducts experiments on up to three datasets. The PSNR is high and achieves state-of-the-art results on the MipNeRF360 dataset.

3. The paper includes an ablation study to analyze the impact of voxel size.

**Weaknesses:**

1. In the abstract and contributions section, the paper claims significant memory savings. However, there is no discussion or validation related to memory consumption throughout the paper.

2. The statements in Table 1 lack justification and contradict the experimental results. It lists that implicit methods are good at rendering quality while explicit methods like SuGaR and 2DGS are not, which contradicts the results in Table 2. Also, stating that implicit methods have low mesh quality is inaccurate, as shown by state-of-the-art methods like Neuralangelo in Table 3.

3. The overall idea is very similar to GOF, which also establishes a transformation from 3D Gaussians to an opacity function. The regularizations are also the same as GOF, borrowed from 2DGS. The paper also adopts MC and MT for mesh extraction. Despite this, the paper does not compare to GOF or discuss it, even though the authors seem aware of it.

4. The mathematical formulation is relatively difficult to understand due to multiple misalignments in definitions. It also lacks sufficient detail on how one equation leads to another.

5. There are insufficient implementation details, making reproduction difficult. It is unclear if neural networks are used, as Figure 1 mentions a "Neural" Gaussian representation.

**Questions:**

1. In Eq (7), how is the integration over the variable $t$ performed? From my understanding, Eq (6) can be used for rasterization by indexing over $i$, but $t$ is a continuous variable along the ray in Eq (7). Do you need sample points on a ray?

2. How do you transform from Eq. (9) to Eq. (11)? Does $\rho(t)$ relate to $\rho(u)$, and if so, how?

3. In L35, the paper states that 2DGS requires a large number of Gaussian primitives, leading to significant GPU memory consumption. This seems to be a common issue in GS-based methods. How is this solved in the paper?

**Limitations:**

The paper does not include a limitation section.

---

> ### Author Rebuttal · Authors · 2024-08-07
>
> > **Q1: Discussion or validation of memory consumption.**
>
> The quantitative comparison of memory usage is already presented in Table 2 of the main text. Compared to explicit GS methods (3DGS, 2DGS), our approach significantly reduces training memory consumption and storage, thanks to the introduction of voxel grids (Section 3.1):
>
> 1. Attributes of 3DGS such as opacity, color, rotation, and scaling are encoded into voxel properties and decoded by several global MLPs, reducing storage requirements.
>
> 2. Voxel grids provide a more structured and clearer GS spatial distribution, saving the number of GS primitives, as shown in Fig 1 in rebuttal file.
>
> 3. Compared to the heuristic growth and pruning strategy of the original 3DGS, our gradient accumulation-based voxel registration strategy controls the growth of 3DGS more effectively, avoiding wasteful distribution of GS in cluttered spaces.
>
> Additionally, we appreciate the suggestion and will include voxel grid ablation studies. Please refer to the response @epfR Q1 for further details.
>
> > **Q2: Statements in Table 1.**
>
> We apologize for any confusion caused by the table and will replace it with the table in @zurJ Q8. Hope this can help to clarify the question occoured on both implicit/explicit methods, as well as our motivation.
>
> > **Q3: Comparison to GOF.**
>
> **Why not compare?**
>
> Our work and the GOF study were developed concurrently. Since GOF was not published at the time of our submission, we did not include it in our quantitative comparison.
>
> **Comparison with GOF.**
>
> Our work is similar to GOF in terms of deriving from ray-gaussian intersections. However, our scene representation is implicit, addressing the common issue of high memory consumption faced by 3DGS. Additionally, we have developed a mapping from opacity to SDF to adapt to general MC and MT algorithms, a consideration lacking in GOF.
>
> **Our Pros:**
>
> 1. As shown in the table, GOF uses explicit GS management, still faces high storage consumption issues, making training large scenes on a single card challenging. Our method achieves better novel view synthesis results with less VRAM usage.
> | Mip 360 | PSNR | SSIM | LPIPS | Storage  |
> | :--- | :---: | :---: |  :---: | :---:  |
> | GOF | 24.53 | 0.733 |  0.245 |  649 M  |
> | GVKF(ours) | **25.47** | **0.757** | **0.240** | **68 M**  |
> 2. GOF requires a long mesh refinement process (sometimes exceeding training time), although it achieves high geometric accuracy, this process is not general for typical MC/MT algorithms. However, our nonlinear mapping can mitigate this issue.
> | Tant | F1 | Meshing Time |
> | :--- | :---: |  :---: |
> | GOF w/ refine | 0.46  |  ~2 h |
> | GOF w/o refine | 0.34 | ~15 min |
> | GVKF(ours) | 0.36   | ~15 min  |
>
> 3. GOF lacks in-depth mathematical analysis, which might confuse volumetric rendering with modified GS rendering. We revisited this rendering approach, providing solid mathematical backing.
>
> **Our Cons:**
>
> While we acknowledge that our current implementation has some geometric precision gaps compared to GOF, the potential reasons include:
> 1. GOF's iterative optimization extraction method achieves more precise isosurfaces.
> 2. As analyzed in @epfR Q1, further adaptation of regularization term to voxel grids might be needed. We leave it as our future work.
>
> > **Q4: Derivation between equations.**
>
> Fig 3 in the rebuttal file demonstrates our difference of volume rendering and 3DGS rendering, corresponding to Eq 6-8.  For Eq 9, we show that the opacity $\rho(t)$ near surface, linearly combined by concentrated kernel functions, can be regarded as a new gaussian distribution. Eq 10-13 to analyze its property after integral (one inflection point, which is extremely similar to Logistic function). Hence we simulate  the nonlinear inverse mapping of opacity to SDF via Logistic function (Eq 14-15).
>
> > **Q5: If neural networks are used**
>
> 3DGS are encoded into one-dimensional vectors and stored in voxel grids for implicit representation. We employ four global MLPs (as described in Equation 2 in the main text) to dynamically decode the necessary attributes of 3DGS during the rendering process.
>
> > **Q6: Question of Eq 7**
>
> We apologize for any confusion caused by our oversight. The correct form of Equation 7 should be:
> $$
> C = \sum_{i=1}^N c_i \cdot \alpha_i \cdot \mathcal{K_i}(0) \prod_{j=1}^{i-1} (1 - \alpha_j \cdot \mathcal{K_j}(0))
> $$
>  We can index the peak of the kernel function for rasterization, which is consistent with the form of origin 3DGS rendering (Eq 6).
>
>
> > **Q7: How do you transform from Eq. (9) to Eq. (11)? Does $\rho(t)$ relate to $\rho(u)$, and if so, how?**
>
> Please note that both $t$ and $u$ represent points along the ray, but they differ in terms of the origin of coordinates. For $t$, the origin is at the camera center, which complicates the analysis of the surface. Therefore, we let $u$ represent a point along the ray, with the origin at the scene surface, as shown in Fig 2 in the main text. From the perspective of the camera, the origin in Fig 2 is at $t_i$.
>
> In Equation 9, due to regularization during the training process, 3DGS near the surface aggregate together to form a new Gaussian distribution $\rho(t)$. However, from the perspective of $t$, the axis of symmetry of $\rho(t)$ is not at the origin. Therefore, we align the axis of symmetry with the origin to obtain $\rho(u)$, as shown in Fig 2. This can be understood as $\rho(t)$ undergoing a simple translation to become $\rho(u)$; the analysis in Equation 10 is entirely based on $\rho(u)$.
>
> > **Q8: How to save Gaussian primitives ?**
>
> Please refer to Q1.
>
> > **Q9: About limitation section.**
>
> Thank you for the suggestion. We will move the limitations section from the appendix to the main text.

---

> ### Comment · Reviewer_6M18 · 2024-08-08
> **Response**
>
> Thank you for your response. I would like to share some of my thoughts on the current draft.
> I have noticed that the paper bears a strong resemblance to Gaussian Opacity Fields (GOF). For instance, as pointed out by Reviewer epfR, Eq. 7 and Eq. 8 closely resemble Eq. 8 and Eq. 9 from the GOF paper. Additionally, the paper incorporates the same regularization as 2DGS (and mentions GOF in L268–L270). Furthermore, the paper adopts MC/MT for meshing, though it lacks sufficient details to ascertain whether the MC/MT implementation is the same as GOF. Based on the table in Q3, it appears that the current meshing implementation is almost identical to GOF w/o refinement (MT is a main contribution from GOF). These observations raise questions about whether the current method was developed concurrently and independently.
>
> Although the method achieves better PSNR with significantly fewer parameters in MipNeRF 360, the author should provide more emphasis on how this gain is achieved, such as the significantly higher PSNR (25.47) achieved with 10x fewer parameters. It should also be carefully checked that there is no test-set pollution.
>
> Moreover, there are several important details missing, such as the settings of MLP and meshing, making reproducibility difficult. Additionally, the presentation is poor, including numerous errors in symbols and tables in the initial submission, which complicates the evaluation process. Despite the author's rebuttal commitment to major revisions, I believe that if a submission requires numerous revisions and further confirmation by the reviewer, it would be better suited for resubmission. Consequently, I maintain my rating for rejecting the submission.
>
>
> **Additional Questions**:
>
> If I understand correctly, then the correct form of Equation 7 is
> $C = \sum_{i}^{N}c_i \cdot \alpha_i \prod_{j=1}^{i-1}(1-\alpha_j)$ since $K(0) = 1$? If so, how is it consistent with the form of origin 3DGS rendering (Eq 6)? Also, should Equation 8 also be corrected?

---

> > ### Author Response · Authors · 2024-08-08
> >
> > > Summarize
> >
> > Thank you for your comments; they will greatly assist us in our future work. As part of our concurrent efforts, we acknowledge that GOF has achieved higher geometric quality in their rendering implementation than we have, and we recognize that this rendering approach was proposed earlier than ours.
> >
> > We will clarify in our paper that equations 5-8 are the main contributions of GOF. However, as pointed out in Q3, while this newly developed rendering method produces attractive results, it is still imperfect: the original GOF paper lacks a mathematical basis and requires extensive post-processing. We hope our analysis can better help readers understand this innovative rendering method and the improvements we have made in Section 3.3.
> >
> > Thank you for your suggestions regarding Section 3.1; we will add more implementation details and provide a more in-depth analysis.
> >
> > We apologize again for the misuse of mathematical formulas in our initial submission, but we assure readers that the revised formulas are correct, and we will thoroughly address your questions.
> >
> > > Answer to Question
> >
> > The equation 7 can be simplified as:
> > $$
> > C=\sum_{i=1}^N=c_i \cdot \alpha_i \cdot \prod_{j=1}^{i-1} (1-\alpha_j)
> > $$
> > Here we keep $\mathcal{K_i}(0)$ for the coherence with equation 5,8. This formulation also appears in the equation 3 of original 3DGS paper. Both equation 7 and equation 6 are valid, the difference is whether consider $\alpha_i$ as constant value when ray-gaussian intersection changes. For perfect aligning with equation 6, the equation 7 could be
> > $$
> > C=\sum_{i=1}^N c_i \cdot \beta_i \frac{\sqrt{k_i}}{\sqrt{\pi}} \cdot \mathcal{K_i}(0) \prod_{j=1}^{i-1}(1-\beta_j \frac{\sqrt{k_j}}{\sqrt{\pi}} \cdot \mathcal{K_j}(0))
> > $$
> > In this scenario, $\beta_i$ is a constant value, and does not change when ray-gaussian intersection changes. More details can be found in next comment.

---

> ### Comment · Reviewer_6M18 · 2024-08-08
> **Additional comments**
>
> After correcting some symbols, I have re-evaluated the submission and would like to share further thoughts on the current formulation. However, some concerns remain that make this paper difficult to accept in its current form.
> 1. **Kernel Function for Ray Intersections**: If a ray intersects with the 3D Gaussian, the kernel function should be expressed as $K_i(t)=G^{2d}\exp⁡(−k_i⋅t^2)$ instead of $K_i(t) = \exp(-k_i \cdot t^2)$. This discrepancy needs clarification. Can the authors explain why the 2D Gaussian factor was omitted, and how this impacts the accuracy of the formulation?
>
>
> 2. **Volume Rendering Formulation (NeRF vs. 3DGS)**: The paper attempts to establish a volume rendering approach using both a density-based (NeRF) formulation and an alpha-based (3DGS) formulation in Eq. 6 and Eq. 7. However, I believe this approach is inaccurate. Referring to Fig. 3 in the rebuttal PDF, if the Gaussian points are treated as sampling points for a continuous version of volume rendering, then the sampling interval between these points should not be omitted in the quadrature of volume rendering. The current formulation suggests that the kernel intersecting with the ray collapses to an infinitesimally small value, but this does not justify **omitting the $\Delta t$ factor**. Therefore, the transformation from Eq. 4 to Eq. 7 seems problematic. Could the authors clarify the rationale behind this omission and how it affects the overall rendering process?
>
>
> 3. **Density Along the Ray**: If the density along the ray is defined as $\rho(t) = \sum_{i}K_i(t - t_i)$, should the second sampling point be represented as $\rho_2(t) = K_1(t_2 - t_1) + K_2(t_2 - t_2)$ instead of just $K_2(t_2−t_2)$? The current formulation seems to *overlook the contribution of preceding Gaussian kernels* when calculating the density at subsequent points. Could the authors provide an explanation for this simplification and its implications on the accuracy of the density calculation?

---

> > ### Author Response · Authors · 2024-08-08
> > **Answer to more Math questions**
> >
> > Thanks for your in-depth analysis, this helps us a lot for the revision.
> > > Q-1 Kernel Function for Ray Intersections
> >
> > In general, the Gaussian kernel function does not take into account the coefficient in front of the exponential, like $e^{-\gamma t^2}$. This means that with different values of $\gamma$, the Gaussian kernel function is not a strict normal distribution. For simplicity, we have retained the general form of the kernel function in our Equation 5. When considering ray gaussian intersection along the ray, the probability density influence of the Gaussian kernel on the ray is a strict normal distribution function $\frac{\sqrt{k_i}}{\sqrt{\pi}}e^{-k_it^2}$. In addition, the Gaussian kernel itself should also have a constant opacity $\beta_i$. To be precise, we have $\alpha_i=\beta_i*\frac{\sqrt{k_i}}{\sqrt{\pi}}$. From the perspective of yours, the missing $G^{2d}$ should be $\frac{\sqrt{k_i}}{\sqrt{\pi}}$, it is already included in $\alpha_i$.
> >
> >
> > > Q-2 Volume Rendering Formulation (NeRF vs. 3DGS)
> >
> > I can mainly understand your question. To clarify, please do not confuse our opacity $\rho(t)$ and volume density $\sigma(t)$ in NeRF. In the main text, we do not use any symbol or concept like $\sigma(t)$.
> >
> >
> > Firstly, we would like to clarify some basic concept for better explanation:
> >
> > **Volume rendering (NeRF)**
> > $$
> > C=\sum_{i=1}^N T_i(1-\exp(-\sigma_i\delta_i))c_i, \quad T_i=\exp(-\sum_{j=1}^{i-1}\sigma_j\delta_j)
> > $$
> > which can be written as:
> > $$
> > C=\sum_{i=1}^{N}T_i \cdot \alpha_i \cdot c_i, \quad   \alpha_i=(1-\exp(-\sigma_i\delta_i)), \quad T_i=\prod_{j=1}^{i-1}(1-\alpha_i)
> > $$
> > These are illustrated in equation 1 and equation 2 of original 3DGS paper.
> >
> > **3DGS rendering**
> >
> > This is the rendering of 3DGS in original 3DGS paper:
> > $$
> > C=\sum_{i=1}^N=c_i \cdot \alpha_i \cdot \prod_{j=1}^{i-1} (1-\alpha_j)
> > $$
> >
> > **Relationship of NeRF and 3DGS rendering**
> >
> > As we can see, the only difference between them is: $N$ in volume rendering is larger. Hence the goal of 3DGS and volume rendering is consistent and the difference is sampling resolution. This is also the fundation of 3DGS rendering equation in original 3DGS paper. We also illustrated this in Fig 3 of rebuttal PDF file.
> >
> > **Please note**
> > 1. The opacity $\alpha_i$ is the function of volume density $\sigma_i$ and sampling interval $\delta_i$ in volume rendering, which is the **accumulated result in a sampling interval**
> > 2. In NeRF, dense sampling is used for better record $\sigma$, rather than opacity $\alpha$
> > 3. The concept of volume density $\sigma$ does not appear in 3DGS and our paper, and rebuttal PDF file. We directly using guassian primitives to represent opacity $\alpha$, rather than volume density $\sigma$
> > 4.  Our goal is to find continuous opacity $\alpha$ along the ray, represented as $\rho(t)$, rather than volume density $\sigma(t)$
> >
> >
> > **Answer your question**
> > - *"Gaussian points are treated as sampling points for a continuous version of volume rendering"*
> >
> > This is right, and as analysed above, this is the fundation of 3DGS rendering equation.
> >
> > - *"The sampling interval between these points should not be omitted in the quadrature of volume rendering"*
> >
> > This is not right, $\alpha_i$ already records the accumulated influence to the ray in a short or long sampling interval, hence $\alpha_i$ should not multiply any $\delta$ again;
> >
> > - *"Why omitting the $\Delta t$ factor"*
> >
> > If we use kernel function to represent continuous volume density $\sigma(t)$, rather than $\rho(t)$ in our paper, the $\Delta t$ should be included to calculate opacity. However, this will make things trouble.
> >
> > > Q3 Density Along the Ray
> >
> > **Contribution of preceding gaussians**
> >
> > In the equation 7, the contribution of preceding gaussian kernels are reflected in the term $\prod_{j=1}^{i-1}(1-\alpha_j \cdot \mathcal{K_j}(0))$.  Hence any kernel before should not be included in current point.
> >
> > **Correct sampling point**
> >
> > Based on equation 5, the second sampling point should be like $K_1+K_2+...+K_N$, the linear combination of all kernel functions, rather than $K_2$ or $K_1+K_2$.
> >
> > **Optimizing kernel fucntions**
> >
> > Based on Equation 5, our continuous opacity function $\rho(t)$ is defined on linear combination of kernel functions, hence the best way to optimize it is to directly optimize the kernel function and linear coefficients, rather than the sampling point represented by all of the kernels. As shown in Equation 7, this way considers every kernel's contribution and the influence of preceding kernels, which is coherent with volume rendering euqation and 3DGS rendering equation in Q2

---

> > > ### Comment · Reviewer_6M18 · 2024-08-10
> > > **Response**
> > >
> > > Thank you for the detailed explanation. While I appreciate the effort, I find that I still struggle to fully grasp some of the equations. I would like to raise the score and lower my confidence level.

---

### Official Review · Reviewer_sv7m · 2024-07-07

**Soundness:** 3
**Presentation:** 3
**Contribution:** 3
**Rating:** 6
**Confidence:** 3

**Summary:**

The paper presents interesting combintation of implicit and explicit representation that achieves efficient and high-fidelity open-scene reconstruction. Basiclaly, they combine the sparse voxel representation attached with per-voxel Gaussian splatting representation and proposes formulation for 3D surface reconstruction and volume rendering.

**Strengths:**

The formulation is mathematically sound.  The paper is well-written. The experiments are thorough and informative. Especially the geometric quality in Figure 5 achieves highly detailed and smooth results compared to baseline.

**Weaknesses:**

The geometric reconstruction is inferior to Neuralangelo, according to Table 3. Can you elaborate more on this? For example, the qualitative results are performed only against the explicit methods. I think the current method is strong enough, but the current paper does not clearly indicate when it fails, which makes the process appear weaker.

Minor comments
- Fonts in figure 2 can be larger.

**Questions:**

- Even though the advantages of the proposed work will be prominent with open scenes, I think the formulation can be applied to general scenes, including objects. How will the performance gap (speed, quality, memory, etc.) be different in different settings? The results indicate that the performance is better for outdoor scenes and comparable to other works for indoor scenes (Table 6). Do you have any intuition what constraints reflect the performance gaps, and why do you emphasize reconstructing open scenes?

- line 121 function -> feature?
- Under Equation 5, $k_i$ is not clearly defined. Can you put definition of the ray-gaussian transfom?
- In equations 6, 7, and 8, is the index $i$ the same as $k$?
- Please make the embedded fonts in Figure 2 larger.
- I do not fully follow the derivation in Equation 11. Why do you put a square $-\rho^2(u)$?
- Figure 6 is hard to interprete. What are the main differences?

**Limitations:**

- The limitations are addressed in apppendix. I think the reconstruction of sky are intentionally blocked in Figure 4. I would appreciate such gap to be revealed - we can interpret the results as a trade-off to achieve more complete scene without holes.

---

> ### Author Rebuttal · Authors · 2024-08-07
>
> > **Q1: The geometric reconstruction is inferior to Neuralangelo as shown in Table 3. Can you elaborate more on this?**
>
> Implicit methods, such as those based on NeRF, typically utilize a global fitting approach for SDF, which allows them to fully leverage the universal approximation capabilities of MLPs. This is advantageous even in areas with sparse viewpoints. However, our current method employs a local line-of-sight-based SDF fitting, a compromise made to adapt to the 3DGS rendering style. This means that regions not covered by the training viewpoints lack fitting capability, resulting in uneven surfaces, as shown in Fig 2 in the rebuttal file. In areas with sparse viewpoint coverage, the distribution of 3DGS is sparse, which hinders the fitting of smooth planes.
>
> Despite this, it is important to note that our method offers more practical advantages compared to implicit representations. For instance, Neuralangelo still relies on computationally intensive volumetric rendering, which results in longer training times (over 24 hours, see Table 3 in the main text). In contrast, our method can be trained in less than 1.5 hours, benefiting significantly from the integration with 3DGS.
>
> We believe that upgrading our current line-of-sight-based implicit representation to a global implicit representation, while avoiding computationally intensive volumetric rendering, is a promising direction. We plan to pursue this as future work.
>
> > **Q2: Fonts in Figure 2 can be larger.**
>
> We will increase the font size in Figure 2 for better readability.
>
> > **Q3: How will the performance (speed, quality, memory, etc.) differ for general scenes versus open scenes? Why performance gaps？**
>
> We observe that current methods based on 3DGS perform adequately for indoor scenes, where there is typically 360-degree viewpoint coverage. However, they underperform in outdoor scenes due to limited viewpoint coverage. Heuristic splitting and pruning strategies in original 3DGS tend to fit the training viewpoints rather than distributing evenly across the space. This leads to poorer novel view synthesis results in outdoor environments. As illustrated in Fig 1 of rebuttal file, without a voxel grid, heuristic GS growth strategies result in uneven spatial distribution of GS, sometimes even creating holes. Conversely, using voxel grids to constrain GS allows for efficient management of their spatial distribution, supporting better novel view synthesis.
>
> Therefore, while our method shows significant improvement in NVS performance for outdoor scenes, the improvement is not as pronounced for indoor scenes. Despite this, the implicit representation using GS consistently saves space across both indoor and outdoor settings:
>
> | Method | Outdoor | Indoor | Avg |
> |--------|---------|--------|-----|
> | GOF (3DGS-based)   | 1045 M     | 254 M    | 649 M |
> | GVKF   | 91 M    | 45 M   | 68 M |
>
> > **Q4: Should "function" be "feature" on line 121?**
>
> Thanks for correct.
>
> > **Q5: Under Equation 5, what is the definition of the ray-Gaussian transform?**
>
> Based on Equation 1 in the main text, the influence of 3DGS $\mathcal{G^{3D}_i}$ in camera space on a one-dimensional ray can be expressed as follows:
> $$ \rho(t) = \exp(-\frac{1}{2}(vt-p)\Sigma^{-1}(vt-p)) $$
> Here, $v$ represents the unit vector of the ray direction. This formula converts the three-dimensional influence of 3DGS into a one-dimensional function along a specific camera ray, which is a one-dimensional Gaussian function. Fig 5 in rebuttal file demonstrates the relationship of this transform. For ease of notation, we express it as:
> $$ \rho(t) = \exp(-k_i \cdot (t-t_i)^2) $$
> where $t_i$ denotes the point along the ray where 3DGS has the maximum impact, also known as the "ray-gaussian intersection," which can be analytically given by:
> $$ t_i = \frac{p^T \Sigma^{-1} v}{v^T \Sigma^{-1} v} $$
>
> For further reference, see:
> *Approximate Differentiable Rendering with Algebraic Surfaces. (ECCV 22)*
>
>
> > **Q6: In equations 6, 7, and 8, is the index $i$ the same as $k$ ?**
>
> We appreciate the feedback. Based on the response by @zurJ Q3, we have corrected the formula and added figure to aid your understanding.
>
> > **Q7: Why is there a square in Equation 11?**
>
> Please note that $\mathcal{T}^\prime(u) = -\rho(u) \mathcal{T}(u)$, as shown in lines L144-L145, is due to the exponential term containing an integral.
>
> > **Q8: Figure 6 is hard to interpret. What are the main differences?**
>
> In Figure 6 of the main paper, we demonstrate the influence of different initial voxel grid resolutions (0.1 to 0.01, 0.001) on the final reconstruction quality. While a finer initial grid (v=0.001) produces higher quality results (see Tab 5), it also results in overly dense distribution of GS points, leading to reduced training speed. As discussed in Section 4.3, the initial voxel grid resolution of v=0.01 yields similar quality in novel view synthesis compared to v=0.001 but with shorter training times. Therefore, we ultimately selected v=0.01 as the initial voxel grid resolution.
>
> > **Q9: The limitations are in the appendix. Can you reveal gaps such as sky reconstruction in Figure 4?**
>
> As you mentioned, our method is designed to reconstruct complete scenes without holes, including distant views and sky, which can be easily cropped or retained as needed. We appreciate the suggestion and will move the limitations section to the main text. Additionally, we will provide more comprehensive scene reconstruction comparisons.

---

> > ### Comment · Reviewer_sv7m · 2024-08-09
> >
> > I read the rebuttal and other reviews. My questions are all answered, which I truly appreciate.

---

### Official Review · Reviewer_epfR · 2024-07-12

**Soundness:** 4
**Presentation:** 4
**Contribution:** 3
**Rating:** 6
**Confidence:** 5

**Summary:**

The paper introduces Gaussian Voxel Kernel Functions (GVKF), a novel approach for efficient 3D surface reconstruction in open scenes, leveraging 3D Gaussian Splatting.

The approach aims to combine the strengths of both implicit representations (NeRFs, Neural SDFs) and explicit representations (2D/3D Gaussian Splatting) to achieve an accurate and fast reconstruction method with low VRAM requirements.

**Details**

1. *Compressing 3D Gaussians Using a Sparse Voxel Grid and MLPs*

The paper proposes optimizing neural 3D Gaussians stored in a sparse voxel grid, reminiscent of Scaffold-GS and Octree-GS. The voxel grid is initialized using the downsampled SfM point cloud output during camera calibration. Each voxel can generate a fixed number of Gaussians, limited to a small range around the voxel. Feature vectors are stored and optimized in the grid, alongside the positions of the Gaussians. To generate the rotation, scaling, opacity, and colors of a Gaussian bound to a voxel, the voxel feature is decoded by an MLP along with the position of the Gaussian. This helps compress the representation while leveraging the natural regularity and fitting power of MLPs. To control the number of Gaussians in the scene, the paper takes inspiration from Scaffold-GS, proposing to subdivide or prune voxels based on gradient accumulation in each voxel.

2. *Defining a Neural Opacity Field with Gaussians to Render Images*

For rendering images, rather than using the vanilla 3DGS formula that relies solely on the opacity of the Gaussians for alpha-blending, the paper proposes explicitly computing the ray-Gaussian intersections and use them to convert Gaussian functions into small kernels. Using these kernels, the paper derives a continuous opacity density function defined along the rays. Similar to Gaussian Opacity Fields (GOF), the implicit opacity field defined with this approach allows for more geometry-accurate rendering and better alignment of the underlying neural Gaussians with the surface of the scene.

3. *Extracting Accurate Surfaces from Neural Opacity Fields*

The paper provides mathematical descriptions of the relationship between the neural opacity field and the underlying surface of the scene, resulting in a formula that maps the opacity function to an SDF. A Marching Cubes algorithm can then be used to extract a surface mesh of the scene.

The authors conducted experiments on several challenging datasets, demonstrating that GVKF achieves high reconstruction quality, real-time rendering speeds, and efficient memory usage, outperforming existing techniques in the literature.

**Strengths:**

1. The paper is well-written and easy to follow.

2. I appreciate the detailed mathematical analysis of the proposed method.

3. The blending formula proposed in the paper allows for more geometry-accurate rendering compared to vanilla 3DGS.

4. The voxel structure proposed in the paper enhances resource efficiency (particularly in terms of VRAM usage) by compressing the 3D Gaussians.

5. The paper includes extensive experiments on multiple datasets to validate the method. The qualitative results (reconstructed surface meshes) are convincing, and the quantitative comparisons effectively demonstrate that the approach achieves better geometric accuracy and rendering quality than several existing works.

**Weaknesses:**

1. The paper is extremely similar to Gaussian Opacity Fields (GOF). The rendering Equation 7 (describing the revisited alpha-blending using an increasing opacity function along rays) and opacity field Equation 8 are almost identical to Equations 8 and 9 from the GOF paper. The difference lies in the generation of Gaussians, as GOF does not use a sparse voxel grid with MLPs to decode Gaussian features. However, GOF achieves better results and shorter optimization times (see Table 1 of the GOF paper). Although the paper mentions GOF, it does not compare with it, even though the paper and code were available five weeks before the deadline. I understand that five weeks is a short period, so GOF could be considered concurrent work rather than preexisting work. However, the results from GOF raise important questions about the claims in the paper: GOF seems to perform better and faster using an almost identical approach, except that GOF does not use a sparse voxel grid representation but just a set of explicit Gaussians. Therefore, it seems the voxel grid is not needed for performance and may be useful only for compressing the representation. As no ablation in the paper compares the VRAM requirement and performance of the approach with or without a voxel grid of neural Gaussians, I would like to hear feedback from the authors about this: Does the voxel grid really help in decreasing the memory consumption of the approach? Doesn’t it just heavily decrease performance?

2. Following my previous point, some important ablations are missing in the paper to better validate several important claims. Specifically, the benefits of the voxel grid are questionable when compared with Gaussian Opacity Fields. Moreover, it is unclear why deriving an SDF from the opacity function is needed for applying Marching Cubes; wouldn’t it be possible to apply MC directly on the opacity field? Does the SDF really improve the quality of the reconstruction?

3. No limitations are provided in the main paper. I encourage the authors to move the limitations from the appendix to the main paper, as this is very important for further research.

**Comment**

I really like the main idea of the paper, as well as the overall approach. However, while being very similar to (but more complicated than) Gaussian Opacity Fields, GVKF achieves lower performance, which makes me question some claims in the paper. I believe the paper needs to better justify why each of its components is needed to support the different claims. I am very interested in hearing feedback from the authors about these points and am willing to increase my rating after more clarifications.

**Questions:**

1. How does the method perform without using a voxel grid? Is there a drop or improvement in rendering quality and surface reconstruction quality? What about the VRAM requirements?

2. Why not apply Marching Cubes directly to the opacity function? Does the SDF really improve the quality of the reconstruction?

**Limitations:**

No limitations are provided in the main paper. I encourage the authors to move the limitations from the appendix to the main paper, as this is very important for further research.

---

> ### Author Rebuttal · Authors · 2024-08-07
>
> > **Q1: How does the method perform without voxel grids? What is the impact on quality and VRAM?**
>
> We conducted an ablation study on the Tant dataset to evaluate the impact of voxel representation and SDF mapping. The results are presented in the table below:
>
> | Ablation |PSNR |F1 | Mem |   Stor | Training Time | Meshing Time |
> | :--- |:---: |:---: | :---: |  :---: |:---: | :---:|
> | GVKF | 26.31 | 0.36 | ~9 G | 90 M | ~1.5 h | ~15 min |
> | w/o voxel | 23.60 (-2.71) | 0.39 (+0.03) | ~16 G (x1.6) | 467 M (x5.2) | ~1.4 h | ~15 min |
> | w/o sdf | 26.31 | 0.30 (-0.06) | ~9 G | 90 M | ~1.5 h | ~15 min |
>
>  **Observations:**
>
> - Utilizing voxel representation significantly improves the PSNR for NVS tasks and reduces memory consumption dramatically compared to the original 3DGS setup. Although there is a slight decrease in the geometric quality of surface reconstruction, we consider this trade-off acceptable.
>
>  **Further Analysis:**
>
> - The motivation behind introducing voxels is to serve as spatial anchors, providing a more regulated, structured distribution of 3DGS, as well as controlled growth and splitting strategies. This is crucial for reconstructing large outdoor scenes and indeed, the introduction of voxels has met our expectations by:
>
>          1. Enhancing NVS performance (refer to Tables 2 and 4 in the main text).
>          2. Reducing memory and storage requirements during training, enabling the training of large scenes on a single GPU.
>
> However, the structured distribution of 3DGS does not significantly improve the results of mesh reconstruction as expected. We hypothesize that the current GS regularization techniques, which are designed specifically for explicit GS in 2DGS, are overly constrained by the voxel grid. The voxel grid restricts the movement of GS within a confined area, which is contrary to the expectation of regularization (to aggregate GS along the same ray path). Without proper regularization, achieving credible geometric quality is challenging. Therefore, we believe it is necessary to adapt the regularization terms to fit the voxel grid framework.
>
> > **Q2: Why not use Marching Cubes directly on the opacity function? Does converting to SDF improve quality?**
>
> As demonstrated in Table Q1 and illustrated in Fig 4 of the rebuttal file, applying MC directly to the opacity field is problematic. We have identified two main reasons for this issue:
> 1. **Linear Assumption of Opacity to SDF Mapping:** Directly using MC requires establishing an isosurface as the surface, which can be considered a linear mapping from opacity to SDF, e.g., SDF(opacity) = -opacity + 0.3 to extract opacity=0.3. This linear assumption does not strictly hold due to the nature of GS distribution.
> 2. **Imprecision Due to Linear Interpolation:** Traditional MC relies on linear interpolation, which, when applied directly, results in inaccurate artifacts due to the non-linear properties of GS distributions, as depicted in Fig 4 of the rebuttal file.
>
> To address these challenges, we propose three viable approaches:
> 1. **Increasing MC Resolution:** This approach reduces the impact of non-linearity but leads to significant storage overhead, especially in open scenes.
> 2. **Iterative Optimization to Approximate the Isosurface:** However, this method incurs substantial computational costs and the optimization algorithm is highly customized, limiting its generality.
> 3. **Finding and Applying the Appropriate Inverse Function as Mentioned in Our Work:** This method incurs almost no additional computational cost and is universally applicable.
>
> Our contemporaneous work, GOF, has adopted the second approach. For further analysis, please refer to the response @6M18.
>
> > **Q3: Limitations are only in the appendix. Why not include them in the main text?**
>
>  Thanks for your suggestion, we will add more analysis of limitations in main text.
>
> > **Q4: GOF results suggest voxel grids might not be necessary for performance. Is the voxel grid needed for memory reduction?**
>
> Please refer to Q1.
>
> > **Q5: Why derive SDF from opacity instead of using MC directly? Does SDF really enhance reconstruction quality?**
>
> Please refer to Q2.

---

> > ### Comment · Reviewer_epfR · 2024-08-12
> >
> > I would like to thank the authors for the detailed rebuttal and very clear answers to my questions.
> >
> > I recommend acceptance for the paper, as I think it brings interesting contributions to the field of mesh reconstruction using 3DGS-based approaches. However, although I appreciate the detailed mathematical framework presented in the paper to support its claims, I think the proposed approach overlaps considerably with the existing work Gaussian Opacity Fields (GOF).
> >
> > Consequently, I decide not to raise my rating, but maintain a Weak Accept.

---

### Official Review · Reviewer_zurJ · 2024-07-13

**Soundness:** 2
**Presentation:** 1
**Contribution:** 2
**Rating:** 4
**Confidence:** 4

**Summary:**

This paper introduces a method for 3D Gaussian reconstruction and surface extraction from the 3D Gaussians. The method can render scenes with low memory usage through a voxel organization. Additionally, the authors derived a continuous implicit field on top of the 3D Gaussians and extracted the surface on this implicit field. Overall, the results look good.

**Strengths:**

(1) The method achieves good reconstruction quality (for both novel view synthesis and surface geometry) with low memory usage.

**Weaknesses:**

(1) Figure 1 is too simple and abstract, lacking intuitive demonstration.

(2) I can somewhat understand what the authors want to do in Equation 5, but could the authors provide some figures for easier understanding?

(3) Equation 6 seems incorrect. There appears to be a mixed use of 'i' and 'k'.

(4) Equation 7 seems incorrect. What does the term 't' mean here? It is not defined or explained here.

(5) Why do we need to convert to an SDF value? I do not understand the reasoning.

(6) I cannot find how the authors convert this implicit field to a surface. Did you use Marching Cubes?

(7) In Equation 15, it seems the final derived SDF is defined on the ray and varies with different view directions. However, to extract a surface, the SDF should not be view-dependent.

(8) I find the writing style of the paper a bit tricky. The decrease in memory usage is inspired by ScaffoldedGS. Regarding the claimed surface reconstruction contribution (as another contribution), although the authors provide many mathematical equations, I struggle to grasp the intuition, and the motivation for these equations is not clearly provided.

Overall, while the results appear promising, I find it difficult to understand how it works, especially considering that several equations are incorrect and lack clear motivations. To me, it feels more like a mathematical wrapper. The storyline is disjointed, lacking intuitive analysis.

Minor issues:
Line 53: We -> we
Line 112: Sfm -> SfM

**Questions:**

(1) In Line 292, what is the unit of “1, 0.1, 0.01, …”?

(2) Is there any visualization results of the voxels? I am curious about its quality.

**Limitations:**

As discussed by the authors in the appendix, the limitations include the inability to reconstruct dynamic scenes and the challenge of distinguishing distant regions without priors. These are interesting for future work to explore.

---

> ### Author Rebuttal · Authors · 2024-08-06
>
> > **Q1: Flowchart is too abstract**
>
> Thank you for your suggestion. We have provided more figures in the rebuttal file to illustrate our pipeline.
>
> > **Q2: The explanation of Eq 5**
>
> As shown in Fig 3 of the rebuttal file, the fourth row shows three 1D Gaussian kernel functions, where $t_i$ denotes the values of $t$ at the peaks of these functions. To represent the continuous function $\rho(t)$, these kernels are linearly combined by $\alpha_i$.
>
> > **Q3: Mixed Use of 𝑖 and 𝑘 in Equation 6**
>
> We apologize for any confusion caused by previous symbol misuse. Below, we clarify and correct the equations in conjunction with Fig 3 in the rebuttal file:
>
> **Equation 6** represents the discrete rendering of original 3DGS:
> $$C=\sum_{i=1}^N c_i \cdot \alpha_i \cdot \mathcal{G_i}^{2D} \prod_{j=1}^{i-1}(1-\alpha_j \cdot \mathcal{G_j}^{2D})$$
> The third row of Fig 3 (rebuttal file) illustrates Gaussian primitive distribution along the ray, showing the collapse of the kernels into a narrow area due to discarding the third row of the 3DGS covariance matrix.
>
> **Adaptation to Kernel Functions:** For 3DGS rendering, we substitute $\mathcal{G_i}^{2D}$ with $\mathcal{K_i}(0)$, leading to
> **Equation 7:**
> $$C=\sum_{i=1}^N c_i \cdot \alpha_i \cdot \mathcal{K_i}(0) \prod_{j=1}^{i-1}(1-\alpha_j \cdot \mathcal{K_j}(0))$$
> This change maintains fidelity to the original rendering equation, optimizing both $\alpha_i$ and $\mathcal{K_i}$ through RGB loss.
>
> **Representation of Scene Surfaces:** Based on the initial definition of $\rho(t)$,
> **Equation 8 is:**
> $$\Phi(t)=\sum_{i=1}^{N}\alpha_i \cdot \mathcal{K_i}(t-t_i) \prod_{j=1}^{i-1}(1-\alpha_j \cdot \mathcal{K_j}(t-t_j))$$
> During training, Equation 7 facilitates rendering without dense sampling. For opacity representation, Equation 8 computes a continuous scene representation along the ray.
>
> > **Q4: The definition of t in Equation 7 is unclear**
>
> Please refer to Q3.
>
> > **Q5: The necessity of converting to SDF values**
>
> As described in Equation 8, our implicit opacity field is defined on the camera ray. This means that to determine the "absolute opacity" at any point Q in space, we must traverse all training cameras to find the minimum value. This process is illustrated in Fig 6 of the rebuttal file, where camera rays may come from various directions. Since $\Phi(t)$ is cumulative and increasing, the minimum value occurs when the corresponding camera ray experiences the least obstruction along its path.
> Once the "absolute opacity" of each point in space is established, we could extract a mesh by isolating an isosurface (e.g., $\Phi(x)=0.3$). However, this approach does not comply with the linear interpolation assumption of the MC/MT algorithms and leads to artifacts, as shown in the Fig 4 of rebuttal file. Therefore, we defined a nonlinear mapping from opacity to SDF to mitigate this issue (Equation 15).
> In our experiments, we used the Marching Tetrahedra (MT) algorithm, although the Marching Cubes (MC) algorithm is also applicable.
> For a more in-depth analysis, please refer to the response @epfR Q-2.
>
> > **Q6,Q7:**
>
>  Please refer to Q5.
>
> > **Q8: The writing style and surface reconstruction contributions are unclear**
>
> We summarize the comparison of 3DGS rendering and volume rendering in the following table, hoping this can clarify our motivation in sec 3.2 of main text. Our goal is to find a new rendering form that combines them.
> | Method           | Math Expression     | Pros                   | Cons                                                 |
> |------------------|---------------------|------------------------|------------------------------------------------------|
> | 3DGS Rendering   | Discrete integration   | Fast rendering         | Hardly fit 3D surfaces due to discrete primitives      |
> | Volume Rendering | Continuous integration | Better 3D surface representation | Low rendering speed due to continuous sampling |
>
> As we know, kernel regression is a non-parametric technique used to estimate the conditional expectation of a random variable. This offers flexibility and adaptability for modeling both continuous and discrete functions.
> Therefore, to  integrate the advantages and mitigate the disadvantages of the above two rendering techniques,
>
> (1) we propose to use multiple kernel functions, which are projected from 3DGS along the ray (see Fig 5 in rebuttal file),  to represent continuous 3D surface;
>
> (2) we take the maximum values of each kernel function to perform discrete numerical integration for rendering;
>
> (3) we use **mathematical equations to prove our proposed rendering strategy is equivalent to 3DGS rendering,** rather than a wrapper. This is illustrated in Fig 3 of rebuttal file.
>
> To summarize, we propose a novel 3d surface reconstruction algorithm, which is represented by continuous function while has fast rendering speed. We hope the provided comparison diagrams in rebuttal file will help clarify our approach.
>
> > **Q9: The unit of voxel size in the ablation experiments**
>
> The dimensions mentioned represent the edge lengths of the voxel grids. The initial 3DGS are constructed using sparse point clouds derived from Structure from Motion (SfM). To accommodate our voxel grid system, we have set up grids of various sizes to filter the sparse point clouds, which also serve as storage containers for the implicit 3DGS.
>
> > **Q10: The visualization effect of voxels**
>
> As shown in the Fig 1 of rebuttal file, voxel grids exhibit a more regular spatial distribution compared to traditional 3DGS. This regularity aids in novel view synthesis and saves storage, representing a significant contribution of our work. For further details, please refer to the response @sv7m Q-1.

---

> ### Comment · Reviewer_zurJ · 2024-08-13
> **Discussion**
>
> First, I want to express my appreciation for your efforts in clarifying the equations and improving the overall clarity of the description. However, I still have concerns regarding the positioning and motivation of the paper.
>
> - My initial question regarding the claimed memory reduction contribution in the abstract remains unaddressed.
> - The statement, 'We take the maximum values of each kernel function to perform discrete numerical integration for rendering,' seems to lead to a somewhat trivial conclusion, which is 'equivalent to Gaussian splatting alpha blending.' I struggle to see this as a novel contribution.
> - I am also confused as to why the authors refer to the method as 'kernel regression.' Kernel regression is not equivalent to 'differentiable rendering with Gaussian kernels.'
> - Furthermore, I personally feel that the experiments are disconnected from the overall narrative of the paper. While the experiments demonstrate that the proposed method outperforms the state-of-the-art (SoTAs), there is a lack of analysis explaining why a 'kernel regression' is necessary and how it enhances performance.
>
> However, I would not insist on rejecting the paper. The performance is excellent, and the experimental results are strong. My concern is primarily with the writing style, which I personally do not favor.
>
>
> As a result, I am inclined to maintain my original score.

---

> ### Author Response · Authors · 2024-08-13
> **Reply to Discussion**
>
> > Discussion-Q1: Memory reduction contribution
>
> The memory reduction is resulted from our proposed voxel grids in Section 1. The voxels serve as spatial anchors, providing a more regulated, structured distribution of 3DGS. They also control the growth and splitting strategies.
> 1. The attributes of 3DGS such as opacity, color, rotation and scaling are encoded into voxel properties and decoded by several global MLPs when rendering, as shown in equation 2 in the main text. This implicit representation greatly saves the memory consumption of 3D gaussian primitives.
> 2. Voxel grids provide more structured and clearer 3DGS distributions in space. While in original 3DGS, the primitives are uncontrollable. As shown in Fig. 1 of our submitted rebuttal file.
> 3. Our gradient-accumulation-based voxel registration strategy controls the growth of 3D gaussian primitives effectively, compared to the heuristic growth and pruning strategy of the original 3DGS.
>
> In addition, the Tab. 2 of main text also demonstrates the efficiency of our method in memory reduction. More in-depth analysis can be found in @epfR Q1, @sv7m Q3
>
> > Discussion-Q2: "Taking the maximum values of each kernel function to perform discrete numerical integration for rendering" is not a novel contribution.
>
> Currently, it appears that direct integration between 3DGS and NeRF is not feasible. If we use 3DGS, it **cannot** represent 3d continuous surface (e.g. SDF function). If we use 3d continuous surface representation, currently it must be optimized through **volume rendering**.
>
> As we mentioned in our response to Q8, our goal is to combine the advantages of 3DGS rendering and the continuous representation in volume rendering. This requires us to solve two problems:
> 1. How can we represent the continuous opacity function $\rho(t)$ on a ray using discrete gaussian primitives?
> 2. How can we optimize this continuous opacity function $\rho(t)$ without using volume rendering?
>
> Our contribution lies in finding a method to solve these two issues:
>
> **Firstly**, we consider the effects of 3D Gaussian primitives on the ray as 1D Gaussian kernel functions. Hence, the weighted sum of $\alpha_i$ of each gaussian primitive can represent the continuous opacity function $\rho(t)$ on the ray. As shown in equation 5 in the main text. This is the defination of kernel regression.
>
> **Secondly**, to optimize the continuous function $\rho(t)$，we take the maximum values of each kernel function (including the coefficient $\alpha_i$) to perform discrete numerical integration for rendering, which is equivalent to 3DGS alpha blending.
>
>
> Please be clarified "equivalent to Gaussian splatting alpha blending" is NOT a trivial conclusion, but demonstrates our proposed method to optimize this continuous function makes sense and is mathematically correct. This indicates we propose a NEW solution, **instead of volume rendering**, to optimize continuous 3D surface.
>
> > Discussion-Q3: Why authors refer to the method as 'kernel regression.' Kernel regression is not equivalent to 'differentiable rendering with Gaussian kernels.'
>
> It is notable that the concept of kernel regression is used to represent continuous functions, rather than to depict the rendering process. Our aim is to represent complex continuous function $\rho(t)$ using a simple set of Gaussian kernel functions. This aligns with the process of kernel regression: for each spatial point, opacity can be calculated through the weighted sum of $\alpha_i$, where the weights come from Gaussian kernel functions.
>
> > Discussion-Q4: Lack of analysis explaining why a 'kernel regression' is necessary and how it enhances performance.
>
> As mentioned in Discussion-Q3, kernel regression is the concept for describing how continuous function $\rho(t)$ formed along the ray. In terms of the reason of improvement, our performance benefits from **continuous representation**.
>
> In the original 3DGS-based methods, such as 2DGS, discrete gaussian primitives are employed to fit surfaces, and mesh extraction relies on TSDF method. This approach may lead to holes in areas with sparse gaussian primitives or occlusions. The produced meshes are often overly smooth, which hinders the representation of fine details. In contrast, with our **continuous representation of kernel regression**, we obtain more details on the surface and have less holes, as shown in Fig. 4 and Fig. 5 of main text as well as our project page.
>
> > Discussion-Summary
>
> We highly value your feedback on our writing style and are dedicated to enhancing the clarity and flow of our paper.
> We sincerely hope the reviewers will recognize our efforts and overall contributions. Thank you once again for your valuable time and constructive feedback.

---

### Author Rebuttal · Authors · 2024-08-07

Dear Reviewers,

Please see the attached PDF page, which includes additional experimental results and formula illustrations, to help clarify our approach. We are deeply grateful for the constructive feedback provided by all reviewers, which has significantly helped improve our paper. We are pleased to receive recognition from reviewers for our paper's:

1. Convincing evaluations and good reconstruction quality (Reviewers zurJ, epfR, sv7m, 6M18).
2. Lower resource consumption (Reviewers zurJ, epfR).
3. Appreciation of our mathematical analysis (Reviewers epfR, sv7m, 6M18).

We have been diligently working to improve the paper in various aspects to address your critiques. Below, we summarize the changes made in the updated draft:
1. Corrected symbol misuse in Equations 6, 7, and 8.
2. Added more intuitive explanations for Equations 5, 6, and 7, with illustrations to aid understanding.
3. Included ablation studies and visualizations for voxel and SDF mapping.
4. Enhanced descriptions of research motivation and implementation details.
5. Moved the limitations section from the appendix to the main text and expanded the explanations.
6. Added comparisons and discussions with GOF.

We will add the following content in the updated appendix:
1. More displays of failure cases.
2. Explanations of the effects in indoor scenes.
3. More detailed derivations (ray-gaussian transform) and illustrations.

Please see our reviewer-specific feedback for more information.

---

### Decision · Program_Chairs · 2024-09-25

**Decision:**

Accept (poster)

**Comment:**

Initially, there were issues with the writing of the paper. The rebuttal was very clear and informative. Reviewer zurJ maintained a borderline reject rating, while the other three reviewers upgraded their ratings to 2x weak accept and 1x borderline accept and agree that there is an interesting contribution and strong results. There was a big discussion on the similarity to GOF. As it was published less than 8 weeks before, it is concurrent work and should not be held against this submission.

Therefore, the decision is to accept the paper. This decision was also discussed with the SAC. The authors are requested to integrate all the feedback to improve the writing of the paper.